# Factors Influencing the Intention to Eat Insects as an Alternative Protein Source: A Sample from Turkey

**DOI:** 10.3390/foods14060984

**Published:** 2025-03-14

**Authors:** Ladan Hajhamidiasl, Merve Nur Uçak, Salim Yılmaz, Murat Baş

**Affiliations:** 1Department of Nutrition and Dietetics, Institute of Health Sciences, Acibadem Mehmet Ali Aydinlar University, Istanbul 34752, Turkey; ladan.hajhamidiasl@live.acibadem.edu.tr (L.H.); murat.bas@acibadem.edu.tr (M.B.); 2Department of Health Management, Faculty of Health Sciences, Acibadem Mehmet Ali Aydinlar University, Istanbul 34752, Turkey; salim.yilmaz@acibadem.edu.tr

**Keywords:** insect-based food, neophobia, sustainability, socio-psychological factors, structural equation modeling

## Abstract

The consumption of insect-based foods has been widely studied in recent years due to their nutritional value and their contribution to sustainability. In this study, the integrated sustainable neophilic insect-based eating model (ISNIEM) was used to investigate the various parameters that influence the intention of members of Turkish society to consume insect-based foods. Structural equation modeling was used to test the ISNIEM. A total of 1194 participants were reached through an online survey. According to the study data, sustainability attitudes (biospheric values, new human interdependence paradigm, attitude toward sustainability, attention to insect welfare) influence individuals’ intentions to consume insect-based foods by interacting with each other; however, intentions do not influence behavior in the same direction. As expected, food neophobia reduced the number of chosen insect-based foods (NCIBF) (β: −0.121; *p* < 0.001), while social norms positively influenced the NCIBF (β: 0.176; *p* < 0.001) and reduced food neophobia (β: −0.307; *p* < 0.001). Meanwhile, social norms and food neophobia did not affect the intention to eat insect-based food (*p* > 0.05). The findings of this study suggest that the ISNIEM may be effective in predicting individuals’ intentions and behaviors toward eating insect-based foods in Turkey.

## 1. Introduction

The global population is growing; however, resources are insufficient to meet this increase. The United Nations predicts that the world population will reach approximately 10 billion by 2050 [1]. In this process, it is thought that the public’s demand for meat consumption will gradually increase; therefore, alternative protein sources may be required to meet this demand [2].

In this context, the inclusion of insect-based foods (IBFs) for human consumption has become a very popular research topic in recent years. The human consumption of some edible insects and their parts was legalized in Regulation 2015/2283 of the European Parliament and of the Council [3].

Raising cows, sheep, and poultry in increasingly larger areas to meet the growing global demand for meat due to population growth is causing significant problems for natural resources and increases greenhouse gas emissions. Studies have shown that insects emit less greenhouse gas and ammonia than cattle and pigs and require less growing space [4].

In addition to being relatively less harmful to the environment, insects are rich in nutrients, containing high levels of protein, vitamin B12, minerals such as iron and zinc, fiber, and w-3 and w-6 fatty acids [5]. One study reported that insects have higher protein and fiber content compared with other animal sources [6]. In addition, insect consumption has been reported to help improve gastrointestinal health due to its fiber and short-chain fatty acid contents [7,8]. IBFs also have antioxidant and anti-inflammatory effects and can even be used as a supplement after exercise due to their rich protein content [7,8].

However, human consumption of these products is crucial to realizing all the environmental and health impacts. While insects have been consumed as food for many years in regions such as Asia (Thailand and China), Africa (Zimbabwe and Uganda), and South America (Mexico and Brasil), their consumption is relatively low in Anatolia, Europe, and the Americas [9]. In Turkey, the prevalence of the consumption of IBFs is also low [10].

Insect production has increased in recent years, and there are currently 310 insect producers worldwide, with new companies being established mainly in Western countries. Insects that are cultivated and consumed as food include the small mealworm (*Alphitobius diaperinus*), the house cricket (*Acheta domesticus*), the yellow mealworm (larvae of the *Tenebrio molitor* beetle), and migratory locusts (*Locusta migratoria*) [11]. The European Union has published a scientific opinion on the consumption of these insects as food, based on the One Health from Farm to Fork principle [12]. They state that the consumption of these products may cause allergic reactions and may pose a health risk due to mycotoxins and pesticide residues, and that caution should be exercised because the scientific literature on the food safety of edible insects is limited [13]. However, it is known that insects produced in controlled farms are safe to eat in the recommended quantities [14,15,16].

Individual attitudes towards the consumption of IBFs are important for this market. Individuals’ attitudes towards the consumption of IBFs vary [17], and analysis of the factors that influence this is very important in guiding this market. Many studies have been conducted to assess consumer attitudes toward IBFs. For example, a study that investigated the attitudes of individuals toward the consumption of edible insects in Turkey found that the vast majority of Turks (94.1%) had never consumed food containing insects, and those who had consumed insects had mostly done so abroad (84%). In general, Turkish consumers were neutral toward the consumption of IBFs, and therefore production and consumption are not common [10]. A systematic review of consumer perceptions and behaviors regarding sustainable protein consumption found that consumers underestimate the environmental impact of meat production and are unlikely to use alternative protein sources [18]. While some research has been carried out into the attitudinal and behavioral factors associated with eating insects, meat consumption and sustainable meat consumption, consumer strategies to reduce meat consumption, and attitudes toward sustainable food consumption, no research has been carried out that directly addresses the attitudinal and behavioral factors associated with eating insects in Turkey [18].

The consumption of IBFs is not widespread in our country, and the number of studies evaluating how individuals view such foods is quite low. Indeed, there are no studies investigating the factors that influence the intention and behavior of Turkish society toward eating IBFs.

The aim of this study was to evaluate the tendency of Turkish people to consume IBFs using the attitude–intention–eating model, including biospheric values and social norms, which are rules or standards of behavior that guide people’s actions, help to create expectations of how others will behave, and promote greater coordination in social life [19].

### 1.1. Literature Review and Hypothesis

Previous studies have investigated the increasing trend towards IBFs and the reasons for this trend. Neophobia, sustainability attitudes, social and cultural norms, knowledge and education levels have been reported to influence individuals’ attitudes towards and acceptability of consuming IBFs [20,21]. Environmental awareness and concern influenced personal norms and the consumption of insect-based diets [22], and concerns about the welfare of animals during the rearing process encouraged the consumption of such diets [23]. As all these factors were examined separately in different studies, it was suggested that all parameters affecting the consumption and attitude of individuals towards such foods should be combined in a single model and consumer behavior should be examined [24].

We aimed to investigate how the following variables affect the attitudes towards and acceptability of consuming IBFs of consumers in Turkey, as also examined by Merlino et al. [24]. We used a mixed approach based on an integrated attitude–intention–eating model (integrated sustainable neophilic insect-based eating model—ISNIEM). This model allows us to assess the influence of socio-psychological, attitudinal, and ethical factors on consumers’ choices of IEIBF. Additionally, we considered the frequency of choosing visual alternatives of insect dishes (visual choice) alongside stated intentions [24]. Thanks to this approach, a significant contribution can be made to the scientific literature on the subject. On the one hand, the study of individual attitudes and behaviors towards different issues (sustainability, animal welfare, etc.) with an integrated approach, associating the consumer with a visual intention criterion, defines a holistic and new model for the study of consumer behavior.

#### 1.1.1. Biospheric Values

Biospheric values are defined as the “concern for nature and the biosphere itself” and those who hold these values prioritize the preservation of nature and respect for the Earth because they believe it is inherently valuable [22]. Individuals with high biospheric values tend to promote environmentally friendly behavior through both individual and environmental self-identity and group identity [25,26]. Furthermore, biospheric values are reported to directly influence the new human interdependence paradigm, which emphasizes the interrelationship between human well-being and environmental protection [24]. Consequently, we hypothesize the following:

**Hypothesis** **1** **(H1).**
*Biospheric values (BVs) have a significant effect on the new human interdependence paradigm (NHIP).*


#### 1.1.2. New Human Interdependence Paradigm

The NHIP recognizes the interdependence between human progress and nature conservation as a dynamic process, where human needs are integrated into natural processes [27]. Social and environmental processes are intertwined, influencing both each other and attitudes towards sustainability [28]. Individuals who believe in the interdependence of humans and nature, viewing humanity as embedded within the natural world, are more likely to be strongly motivated to protect the environment [29]. Consequently, we hypothesize the following:

**Hypothesis** **2** **(H2).**
*The new human interdependence paradigm (NHIP) has a significant effect on the attitude toward sustainability (ATS).*


#### 1.1.3. Attitude Toward Sustainability

Sustainability is defined as “meeting the needs of the present without compromising the ability of future generations to meet their own needs” [30]. Individuals’ attitudes towards sustainable consumption are positively related to sustainable behavior [31]. Based on this, we hypothesize that individuals who are more concerned with sustainability may also be more sensitive to the ethical considerations surrounding insect consumption. Consequently, we hypothesize the following:

**Hypothesis** **3** **(H3).**
*The attitude toward sustainability (ATS) has a significant effect on the attention to insect welfare (AIW).*


#### 1.1.4. Attention to Insect Welfare

Respect for the welfare of insects does not imply that they should not be used in food production. However, it requires the application of a consistent set of ethical standards in their farming practices [32]. While there is no universally accepted definition of animal welfare, and consumer opinions regarding insect welfare remain underexplored [33], ethical and environmental motivations have been identified as strong drivers of food choices [34]. Consequently, we hypothesize the following:

**Hypothesis** **4** **(H4).**
*The attention to insect welfare (AIW) has a significant effect on the intention to eat insect-based foods (IEIBF).*


#### 1.1.5. Intention to Eat Insect-Based Foods

Despite the fact that the protein, fat, mineral, and amino acid profiles of insects confirm their status as a good source of nutrients [35,36], the habit of eating IBFs remains limited, with the exception of certain specific and local cases. Factors influencing the intention to consume IBFs are diverse, including environmental, ethical, health, social, cultural, and psychological factors [37]. While it is well known that intentions do not always translate into actual behavior, this study aims to further investigate whether the intention to eat insect-based foods (IEIBF) leads to actual food choices, as measured by the number of chosen insect-based foods (NCIBF). Consequently, we hypothesize the following:

**Hypothesis** **5** **(H5).**
*The intention to eat insect-based foods (IEIBF) has a significant effect on the number of chosen insect-based foods (NCIBF).*


#### 1.1.6. Food Neophobia

Neophobia (the tendency to avoid unfamiliar foods) is one of the main factors influencing the acceptability of IBFs [38]. It is expected that individuals exhibiting neophobic tendencies will be more reluctant to consume IBFs [39,40,41]. Similarly, a study conducted in our country revealed that consumers’ neophobia attitudes have a significant and negative impact on their behavioral intentions towards the consumption of edible insects [42]. Consequently, we hypothesize the following:

**Hypothesis** **6a** **(H6a).**
*Food neophobia (FNEO) has a significant effect on the intention to eat insect-based foods (IEIBF).*


**Hypothesis** **6b** **(H6b).**
*Food neophobia (FNEO) has a significant effect on the number of chosen insect-based foods (NCIBF).*


#### 1.1.7. Social Norms

Social influence is the process by which an individual’s behavior is altered due to the presence or actions of others [43]. Consumer psychology indicates that the behavior of those around an individual, or even the knowledge of their actions, can significantly impact consumption and purchasing decisions [44]. The willingness to consume insect-based protein is particularly influenced by social norms [45]. Research has shown that people are more likely to consume insects in group settings and observing others eating insects may encourage others to try IBFs [43]. However, social influence is not always positive; individuals’ choices and purchasing behaviors can also be shaped by exposure to negative views. Specifically, awareness of the community’s negative opinions about insect consumption may act as a barrier to acceptance of these foods [46]. Consequently, we hypothesize the following:

**Hypothesis** **7a** **(H7a).**
*Social norms (SNs) have a significant effect on the intention to eat insect-based foods (IEIBF).*


**Hypothesis** **7b** **(H7b).**
*Social norms (SNs) have a significant effect on the number of chosen insect-based foods (NCIBF).*


**Hypothesis** **7c** **(H7c).**
*Social norms (SNs) have a significant effect on food neophobia (FNEO).*


The theoretical model thus integrates environmental, psychological and social dimensions, providing a comprehensive framework for understanding individuals’ IBF consumption behavior. The items used in the model to assess these parameters are detailed in Table 1.

## 2. Materials and Methods

The theoretical model used in this study is based on established psychological and environmental behavior theories that aim to explain individuals’ preferences for insect-based foods (IBFs) through interconnected structures.

Figure 1 presents the theoretical model. It visualizes a series of hypotheses explaining individuals’ preferences for insect-based foods. In the model, the biospheric values (BVs) variable represents individuals’ environmental sensitivity, and the effect of these values on the new human interdependence paradigm (NHIP) is examined (H1). The NHIP influences individuals’ attitudes toward sustainability (attitude toward sustainability—ATS) (H2), and these attitudes, in turn, impact the attention given to insect welfare (attention to insect welfare—AIW) (H3). AIW affects individuals’ intention to consume insect-based foods (intention to eat insect-based foods—IEIBF) (H4), which subsequently influences the number of chosen insect-based foods (number of chosen insect-based foods—NCIBF) (H5). Additionally, the effects of food neophobia (FNEO) and social norms (SNs) on both IEIBF and NCIBF (H6a, H6b, H7a, H7b), as well as the effect of SN on FNEO (H7c), are explored. In the top-right corner, control variables such as “Age and Sex” are displayed within a separate box.

### 2.1. Participants

Participants were recruited via social media, supplemented by snowball sampling. A 5:1 subject-to-item ratio was used to calculate the sample size [47]. Accordingly, the minimum sample size for the 68 items was 340 participants. In the study in which the questionnaire was used for the first time, the number of participants was 1402 [24]. In this study, a total of 1194 participants completed the online questionnaire.

The target population of this study includes individuals from the general population with varying attitudes and behaviors toward IBFs. Participants were recruited via social media, supplemented by snowball sampling, to ensure a diverse sample. This approach enabled us to reach individuals across a broad age range (18–79 years, mean = 38.4 ± 12.48) and gender distribution (68.9% female). Although the sample included a higher proportion of women, this is consistent with previous research suggesting that women are more likely to engage with sustainability-related topics [48].

Each participant voluntarily agreed to participate in this study and signed an informed consent form. Data were collected online via a questionnaire. At the beginning of the questionnaire, participants were provided with a detailed informational text about concepts such as biospheric values and sustainability, ensuring they were informed about these concepts and the study process.

The study protocol was approved by the Acibadem University Medical Research Ethics Committee (2024-11/484). This study was conducted according to the tenets of the Declaration of Helsinki.

### 2.2. Data Collection Procedure

A structured questionnaire previously developed by Merlino et al., which we translated into Turkish, was used to determine the factors influencing participants’ intentions, attitudes, and behaviors toward consuming IBFs [24]. These questions were chosen because they are the most comprehensive survey developed in this field and because the food cultures of Italy and Turkey are similar to each other, although there are small differences. Moreover, cultural adaptation was ensured during translation process.

The questionnaire consists of 30 questions and 68 items in total. Questions 1–10 ask for demographic information; questions 11–13 ask about the use of environmentally friendly products. From the 14th question, “factors influencing the intention to consume insects” are analyzed as follows: Question 14, BVs [49], includes items on the protection of the environment and the biosphere; Question 15, the NHIP [27], assesses the relationship between human progress and nature conservation; Question 16, ATS [24,50], draws attention to the environmental impact of the food chain and to social and economic sustainability issues; Question 17, SNs [51], examines the influence of local norms on behavior; Question 18, IEIBF [24], includes individuals’ attitudes toward consuming IBFs; Question 19 assesses the perceived risks of eating IBFs; Question 20 assesses the participant’s attitude toward insect welfare [33]; and Question 21 asks about the individual’s intention to purchase and consume IBFs. Questions 22–29 ask about the status of consumption of IBFs by means of photography [24]. Finally, in Question 30, the attitudes of individuals toward new and different types of food are assessed using the FNEO Scale [33,52], which has been validated and is reliable in our country. The parameters assessed are detailed in Table 1.

#### FNEO Scale

The FNEO Scale was developed by Pliner and Hobden [33,52] in 1992 to assess FNEO, defined as the aversion to and/or avoidance of eating new foods. The original scale, which contains 10 items, 5 positive and 5 negative, is scored on a 7-point Likert scale ranging from “strongly disagree” to “strongly agree”. The Turkish version of this scale was created by Duman et al. [33,52] in 2020. As for the original scale, the validity and reliability of the 10 items on the Turkish scale have been confirmed.

### 2.3. Statistical Analysis

Data analysis was performed through coding using the lavaan, psych, semTools, ggplot2, and GGally diagrammer packages in R 4.3.1 [53,54,55,56,57]. Normality was assessed according to whether the values of skewness and kurtosis were between 1 and +1. Exploratory factor analysis (EFA) was conducted using Maximum Likelihood (ML) and Principal Axis Factoring (PAF) methods to determine the most appropriate factor structure. The adequacy of the data for factor analysis was evaluated through the Kaiser–Meyer–Olkin (KMO) test and Bartlett’s Test of Sphericity. When analyzing the validity of the latent variables formed by the observed variables, factor loadings and the average variance explained were interpreted. In addition, the composite reliability, Cronbach alpha, and Kuder–Richardson 20 values were used in the reliability analysis. To evaluate the potential presence of Common Method Bias (CMB), the Harman Single-Factor Test was conducted. Scatter plot analyses were performed to visualize the relationships between the independent variables and the dependent variable (NCIBF), revealing non-linear patterns in some variables. To confirm the presence of non-linearity, the Ramsey RESET test was conducted (RESET = 4.3489; *p* < 0.001), demonstrating that the model suffered from a missing functional specification and required additional terms to address non-linearity. Based on these results, polynomial regression models up to the second degree were tested and included in the SEM to appropriately capture the non-linear effects of certain variables. First-degree and second-degree polynomial terms were added for variables identified as non-linear, ensuring a more accurate representation of their relationships. For Confirmatory Factor Analysis (CFA) within the structural equation modeling (SEM) framework, the Maximum Likelihood with Robust Standard Errors (MLR) estimator was used to account for potential non-normality and heteroscedasticity. Similarly, the SEM model was estimated using MLR, ensuring robust parameter estimation under non-normality and heteroscedasticity. The model fit was assessed using chi-square to degrees of freedom ratio (χ^2^/df), root mean square error of approximation (RMSEA), standardized root mean square residual (sRMR), comparative fit index (CFI), and the Tucker–Lewis index (TLI). Spearman and Pearson correlations were used for the correlational analyses. Analyses were performed with 95% confidence intervals (*p* < 0.05).

## 3. Results

The characteristics of the 1194 people included in this study are shown in Table 2. The majority (68.9%; n = 824) of the participants were women, and the mean age was 38.4 ± 12.48 years. The results showed that 49.3% of the participants (n = 590) had a bachelor’s degree, and 22.4% of the participants (n = 238) had an income between 20,001 and 30,000 Turkish liras, which is below the poverty line. The results also showed that 40.6% of the participants (n = 486) paid some attention to green products. While 43.7% of the participants reported that they paid attention to the sustainability of the products they purchased, 49.4% paid attention to the health benefits of products (Table 2).

Table 3 presents the results of the exploratory factor analysis (EFA), Kaiser–Meyer–Olkin (KMO) values and Bartlett’s Test of Sphericity. Both Maximum Likelihood (ML) and Principal Axis Factoring (PAF) methods were tested to determine the most appropriate extraction method. The suitability of the single-factor structure was assessed using the Eigenvalues, Scree Plot, and Parallel Analysis, ensuring that a one-factor solution adequately represents the data.

The exploratory factor analysis results indicate that all factors demonstrate adequate construct validity, with SS Loadings ranging from 2.29 to 5.96 and explained variance values between 53% (SNs) and 88% (BVs), suggesting strong factor structures overall. KMO values are above the acceptable threshold (≥0.72), confirming sample adequacy, while Bartlett’s test results (*p* < 0.001) support the suitability of factor analysis. Factor loadings are generally high (≥0.70), indicating strong associations between items and their respective factors, particularly for BV (0.94–0.96), NHIP (0.88–0.92), and IEIBF (0.97–0.98). While SNs exhibit the lowest explained variance (53%), they remain within the acceptable range for factor retention, though its relatively weaker loadings (0.67–0.76) suggest that further refinement may enhance its robustness (Table 3).

Table 4 provides a detailed overview of the Confirmatory Factor Analysis (CFA) factor loadings, composite reliability scores, and total variance explained for all latent variables. The results demonstrate that all item loadings were above the acceptable threshold of 0.5, with composite reliabilities ranging between 0.8 and 0.9, indicating strong internal consistency. Additionally, the alpha value for the number of insect-based foods selected was 0.925, with a Kuder–Richardson 20 reliability value of 0.929, confirming the reliability of the binary scoring method. These results further emphasize the robustness of the scales’ psychometric properties. Specifically, the unidimensionality of the latent variables was strongly supported by high Omega Hierarchical (ωH) values, ranging from 0.85 to 0.97 across all factors. The latent variables biospheric values, NHIP, and ATS demonstrated particularly strong unidimensionality, with ωH values of 0.96 or above, indicating that a single general factor accounts for the majority of variance. While the SN and FNEO scales showed slightly lower ωH values (0.85 and 0.87, respectively), these were still within acceptable thresholds, affirming their validity. Furthermore, discriminant validity was confirmed using the Fornell–Larcker criterion, as the square root of the Average Variance Extracted (AVE) for each latent variable exceeded its correlations with other variables. For instance, the AVE square root values ranged from 0.71 to 0.94 across factors, consistently surpassing the highest inter-factor correlations. These findings indicate that each latent variable is distinct and measures a unique construct, further strengthening the validity of the scales. Furthermore, discriminant validity was confirmed using the Fornell–Larcker criterion, as the square root of the Average Variance Extracted (AVE) for each latent variable exceeded its correlations with other variables. For instance, the AVE square root values ranged from 0.71 to 0.94 across factors, consistently surpassing the highest inter-factor correlations. These findings indicate that each latent variable is distinct and measures a unique construct, further strengthening the validity of the scales.

The bivariate correlations between the factor variables are shown in Table 4. Significant positive relationships were found between BVs and NHIP (r = 0.668), ATS (r = 0.599), AIW (r = 0.167), SNs (r = 0.196), and IEIBF (r = 0.190), and a significant negative relationship was found between BVs and FNEO (r = −0.143). There were significant positive relationships between NHIP and ATS (r = 0.641), AIW (r = 0.182), SNs (r = 0.233), and IEIBF (r = 0.144) and significant negative relationships between NHIP and FNEO (r = −0.204). ATS showed a significant positive relationship with AIW (r = 0.177), SNs (r = 0.329), and IEIBF (r = 0.169), while showing a significant negative relationship with FNEO (r = −0.232). AIW showed a significant positive relationship with IEIBF (r = 0.212) and a significant negative relationship with FNEO (r = −0.132). Similarly, SNs showed a significant negative relationship with FNEO (r = −0.238) and a significant positive relationship with NCIBF (r = 0.180). There were also significant negative relationships between IEIBF and FNEO (r = −0.070), IEIBF and NCIBF (r = −0.159), and between FNEO and NCIBF (r = −0.237) (Table 5).

In the CFA conducted prior to the SEM model, covariances were defined within the same factor in accordance with theoretical assumptions to account for shared variances among the relevant indicators. The included covariances were as follows: NCIBF3~NCIBF4 (r = 0.702), NCIBF1~NCIBF2 (r = 0.536), NCIBF5~NCIBF7 (r = 0.526), SN1~SN3 (r = 0.475), NCIBF6~NCIBF8 (r = 0.420), SN4~SN5 (r = 0.420), and ATS4~ATS5 (r = 0.411). All these covariances were statistically significant (*p* < 0.001), ranging from moderate to high, indicating meaningful interrelations among the selected indicators. Additionally, the factor loadings for the variables were generally strong, supporting the validity of the measurement model. For instance, items associated with biospheric values (BVs) had high standardized factor loadings ranging from 0.929 to 0.959, while those related to attention to insect welfare (AIW) showed loadings between 0.772 and 0.942, highlighting the robustness of these constructs. However, some items, such as those under social norms (SNs) and intention to eat insect-based foods (NCIBF), had slightly lower loadings (e.g., 0.611 and 0.662, respectively) but remained within acceptable limits. These results demonstrated a strong alignment between observed variables and their corresponding latent constructs, enhancing the reliability and interpretability of the CFA findings (Figure 2). Additionally, the fit indices results are provided in Table 6.

All of the fit indices presented meet or exceed the commonly accepted thresholds, indicating a strong overall model fit. Specifically, the χ^2^/df value of 4.07 falls within the recommended limit of ≤5.0, while the CFI (0.936) and TLI (0.930), along with their robust counterparts (Robust CFI = 0.939, Robust TLI = 0.933), all surpass the 0.90 threshold, demonstrating good incremental fit. The RMSEA (0.051) and Robust RMSEA (0.056) remain below the 0.08 cutoff, further supporting model adequacy. Additionally, the SRMR of 0.055 satisfies the recommended criterion of <0.08, ensuring acceptable residual-based fit. The GFI (0.922) and AGFI (0.914) also lie within the acceptable range of 0.85 to 1.00, reinforcing the model’s goodness-of-fit. In the Harman Single-Factor Test, a single factor explained 28% of the variance (SS Loadings: 11.76), confirming that common method bias is not a serious concern, as it remains well below the 50% threshold (Table 6).

In Figure 3, a scatter plot analysis was conducted to visualize the relationships between NCIBF and the independent variables. As a result of this analysis, some variables were found to be non-linear. To account for these non-linear relationships, a polynomial regression model was established. Additionally, the Ramsey RESET test was applied to check for linearity, and the test results (RESET: 4.3489; *p* < 0.001) indicated that the model suffers from a missing functional specification, suggesting that non-linear effects should not be ignored. Furthermore, the presence of multicollinearity in the model was examined using Variance Inflation Factor (VIF) analysis. Blue is the linear regression curve and red is the loess regression curve (Figure 3).

According to the VIF results presented in the table, there is no serious issue of multicollinearity, as all variables have VIF values below 5. In particular, AIW (1.087773), IEIBF (1.095272), Age (1.131385), and SNs (1.171359) have very low VIF values, indicating that there is no strong linear relationship among the independent variables. The highest VIF values are observed for NHIP (2.518469) and BVs (2.290271); however, these values remain within an acceptable range, suggesting that the model does not face a significant risk of multicollinearity (Table 7).

Based on the polynomial model up to the second degree, which was established after the scatter plot analysis while controlling for age and sex linearly, it was determined that the ATS, FNEO, SN, and IEIBF variables are non-linear. ATS (*p* = 0.0048) and IEIBF (*p* < 0.001) were included in the model as first-degree polynomial terms, while FNEO (*p* < 0.001) was found to be significant as a second-degree polynomial, leading to the addition of a quadratic term to capture its non-linear effect. Similarly, the SN variable was incorporated into the model as a first-degree polynomial and exhibited a significant effect (*p* < 0.001). Identifying and appropriately incorporating these non-linear variables into the model was necessary to more accurately capture their impact on NCIBF (Table 8).

The structural equation model reveals significant relationships between sustainability attitudes, food neophobia, and insect-based food consumption intentions, with key insights from both the theoretical model and modifications. Biospheric values (BVs) strongly predict the new human interdependence paradigm (NHIP) (β = 0.702, *p* < 0.001), suggesting that individuals with higher ecological values are more likely to understand the interdependence between human progress and ecological balance. Sex also plays a role, with males slightly more likely to internalize the NHIP (β = 0.058, *p* = 0.008), while age was found non-significant and removed in later iterations. The NHIP positively influences attitudes toward sustainability (ATS_poly1) (β = 0.405, *p* < 0.001), with older individuals (β = 0.094, *p* < 0.001) and males (β = 0.057, *p* = 0.017) also exhibiting slightly stronger sustainability attitudes. These attitudes, in turn, positively affect attention to insect welfare (AIW) (β = 0.098, *p* = 0.009), although males show significantly lower attention to insect welfare (β = −0.056, *p* < 0.001) compared to females, and age has no significant effect in this context. Regarding food neophobia (FNEO_poly2), stronger social norms (SN_poly1) significantly reduce resistance to trying unfamiliar foods (β = −0.233, *p* < 0.001), while older individuals are more likely to exhibit neophobia (β = 0.262, *p* < 0.001) and males less so (β = −0.071, *p* = 0.014). Interestingly, while attention to insect welfare (AIW) positively predicts the intention to eat insect-based foods (IEIBF_poly1) (β = 0.178, *p* < 0.001), other factors such as age, sex, and food neophobia were found non-significant and removed in the modified model. The modifications further highlight new relationships, such as the significant influence of social norms on sustainability attitudes (β = 0.199, *p* < 0.001) and the NHIP on attention to insect welfare (β = 0.160, *p* < 0.001), emphasizing the interconnected nature of these constructs. Additionally, covariances were included to capture nuanced relationships, such as the negative associations between the NHIP and food neophobia (β = −0.147, *p* < 0.001) and between the NHIP and insect-based food intentions (β = −0.127, *p* = 0.005), suggesting that individuals with stronger ecological values may be less neophobic but cautious about IBF consumption (Table 9).

In the base model of the structural equation model, controlling for age and gender, the alternative hypothesis H7a (SNs have a significant effect on the IEIBF) was rejected (standardized β: 0.056; *p* = 0.055). The alternative hypothesis H6a (FNEO has a significant effect on the IEIBF) was also rejected (standardized β: −0.015; *p* = 0.612). After removing the effects of these variables from the model, the goodness of fit indices were as follows: RMSEA: 0.054 (0.040–0.069); sRMR: 0.032; df/χ^2^: 4492, *p* < 0.001; CFI: 0.975; TLI: 0.924. When all variables in the model were controlled for age and gender, BVs had a significant positive effect on the NHIP; therefore, H1 was supported (standardized β: 0.717; *p* < 0.001). The alternative hypothesis (H2) regarding the effect of the NHIP on ATS was also supported, and the NHIP had a strong and positive significant effect on ATS (standardized β: 0.985; *p* < 0.001). Similarly, H3, regarding the effect of ATS on AIW, was supported, and ATS had a positive and significant effect on AIW (standardized β: 0.338; *p* < 0.001). H4, regarding the effect of AIW on the IEIBF, was also supported with a strong positive effect (standardized β: 0.968; *p* < 0.001). The effect of IEIBF on NCIBF (H5) was also significant, but the effect was negative (standardized β: −0.157; *p* < 0.001). The hypothesis regarding the effect of FNEO on the NCIBF (H6b) was also supported (standardized β: −0.121; *p* < 0.001), and this effect was also negative. Hypotheses H7b and H7c, regarding the effects of SN on both the NCIBF and FNEO were significant (standardized β: 0.176; *p* < 0.001 for NCIBF; standardized β: −0.307; *p* < 0.001). SNs had a positive effect on NCIBF and a negative effect on FNEO (Figure 4).

The structural equation model demonstrated a strong fit to the data based on multiple fit indices. Red lines denote negative effects, blue lines denote the theoretical non-model unexpected lines, and green lines denote positive effects. Dotted lines indicate insignificant effects. The χ^2^/df ratio was 3.06, indicating an acceptable model fit. The CFI (0.967), TLI (0.914), and their robust versions (0.968 and 0.916, respectively) all exceeded the recommended threshold of 0.90, supporting good comparative model fit. The RMSEA (0.066) and sRMR (0.045) were below the 0.08 cut-off, indicating low residual error. Additionally, the Goodness-of-Fit Index (GFI = 0.961) and Adjusted Goodness-of-Fit Index (AGFI = 0.985) confirmed strong overall model adequacy. These results suggest that the model provides a satisfactory representation of the data and aligns well with theoretical expectations (Table 10).

## 4. Discussion

Although Turkish people do not commonly consume IBFs, insect consumption is more prevalent in regions such as the Far East and Africa [9]. The consumption of IBFs has been an important research topic in Europe in recent years, mainly due to their nutritional and environmental benefits. Several market studies have been conducted in European countries such as Spain [38], Poland [58], Hungary [59], and Switzerland [60] to assess consumer attitudes toward the consumption of IBFs. In these studies, EFSA-approved insect species have played a critical role in consumer acceptance by adhering to strict human health and safety regulations. These regulations have alleviated consumer concerns and contributed to increasing consumption.

Although the number of studies conducted within this field in Turkey is low, it has been reported that the majority of people do not consider insects an edible food, and some are not inclined to consume insects as protein alternatives due to religious beliefs [61]. Since religion plays a significant role in Turkish society, dietary laws and religious perceptions may strongly influence the acceptance of IBFs. Certain beliefs may lead individuals to perceive insects as impure or unsuitable for consumption, further reinforcing their reluctance to consider them as a viable food source [10]. A market study conducted in Cappadocia found that restaurateurs believed that the presence of insect products on their menus would not contribute to tourism [62]. There appears to be little incentive for people in the region to produce IBFs.

Our results suggest that individuals’ consumption of IBFs follows three main distinct pathways.

The first pathway of the model is a sequential chain of sustainability attitudes starting with biosphere value. The strong and positive effect of BVs on the NHIP suggests that environmentally conscious individuals are more likely to believe in and embrace the human-nature connection. The strong effect of NHIP on ATS suggests that people who support the human–nature connection have more positive attitudes towards sustainability issues. Positive ATS have a positive effect on AIW, but this effect is moderate and weaker than other relationships. AIW has a strong and positive effect on IEIBF, suggesting that ethical concerns positively influence consumption intentions. Attitudes towards sustainability show to increase attention to animal welfare, which in turn increases intention to consume. However, age, gender and neophobia did not affect individuals’ intention to consume IBFs. This suggests that, in addition to certain demographic factors, environmental attitudes and ethical values may play a more decisive role in shaping consumption decisions.

People with awareness in this area are more open to eating insects. In a study conducted in Spain by Ros-Baro et al., awareness of the impact of IBFs consumption on sustainability increased the intention to consume these foods [38]. However, intentions to eat should not be taken to mean that people will consume these foods. According to the results of this study, a high IEIBF did not increase the NCIBF at the same rate; on the contrary, it decreased them. Mancini et al. reported that individuals with low FNEO believed they could consume IBFs but did not actually consume them [37]. Therefore, it is not possible to say that intention directly influences behavior. Failure to translate intentions into behavior may be due to perceived risks or other barriers, such as psychological and cultural factors. Individuals may experience food neophobia, making them reluctant to try novel foods such as IBFs, or they may associate these products with disgust due to ingrained cultural norms. In addition, a lack of social acceptance and familiarity may reinforce resistance and prevent consumers from adopting IBFs despite their positive intentions. To bridge the gap between intention and behavior, targeted educational campaigns can increase awareness of the benefits of IBF, while sensory exposure and product integration into familiar dishes can increase acceptance. In addition, leveraging social norms through endorsements and policy incentives can help normalize consumption and reduce psychological barriers.

Previous consumption of IBFs and familiarity with these foods are known to positively influence the intention to consume. In a European study that included 887 participants, experience with IBFs influenced general attitudes and subsequent willingness to purchase IBFs [63]. In this study, almost 90% of participants had never consumed IBFs, which may explain the low consumption compared to intentions. Providing information on the environmental and health impacts of insects and incorporating insects into recipes as part of the community’s eating habits will help to familiarize consumers with these foods and increase their availability [38,64].

The other two pathways that influence individuals’ consumption of IBFs are the influence of neophobia and social norms on consumption. The influence of SNs on FNEO is positive but weak. This means that the social norms of the environment slightly increase individuals’ fear of IBFs.

The effect of food FNEO on the NCIBF was negative. This means that as fear of IBFs increases, the number of choices of such foods decreases. This fear can stem from psychological factors, such as neophobia and disgust sensitivity; biological factors, including innate aversions to potentially harmful foods; and social influences, where cultural norms and societal attitudes shape individuals’ reluctance to consume IBFs [41]. Similarly, a study has reported that social or personal rejection of IBFs progresses in parallel with neophobia [48].

When analyzing the parameters that influence people’s attitudes toward IBFs, the most important factor was reported to be FNEO [20,37,58,63,65,66]. The desire to eat insects is reduced when FNEO is high [65,66]. In a study of 181 children in Denmark, neophobia was reported to have a significant impact on the willingness to consume IBFs. People who are unaware of the impact of insects on health and the environment and who are not open to eating new foods will not even want to try such foods [65]. According to the data obtained in this study, the fact that neophobia reduced the number of insects that individuals would choose to consume supports the findings of the previous literature.

A review of consumer perceptions found that sustainability factors influence insect consumption, but that the main factor was disgust, in addition to neophobia [20]. Even the Chinese, who have a long history of eating IBFs, have been reported to have phobias and feelings of disgust toward certain types of insects, which affect their purchase of IBFs [67]. A Polish study which examined parameters such as attitudes toward the acceptance of insects as meat substitutes and as food, attitudes toward the health effects of food, individuals’ attitudes toward the environmental effects of food choices, FNEO, disgust sensitivity, and variety seeking reported that there was a significant correlation between FNEO and disgust sensitivity and that individuals with these characteristics were less likely to accept IBFs [58].

To reduce neophobia, people should be informed about both the health and environmental benefits of IBFs. Studies have shown that attitudes toward the consumption of IBFs are positively influenced after an informative seminar [37]. A study of 7222 participants from the United States and Europe found that people in these regions had very low levels of knowledge about IBFs and their health effects and generally had neither positive nor negative attitudes toward consuming them [68]. A Hungarian study reported that, in Hungary, the public is less health-conscious than in Belgium but is more concerned about the environmental impact of food choices [69]. Different marketing strategies should therefore be developed to increase the use of insects as a food source, in addition to information to reduce neophobia [66]. In addition, social media campaigns and collaborations with influential figures in the sustainability movement can successfully increase consumer interest and acceptance of insect-based products.

In addition to the main pathways specified in the model, we discovered new pathways that go beyond the theoretical model as a result of the modifications made. The newly developed relationships represent previously unexpected and potentially significant links within the model.

Among these newly discovered pathways, SNs and BVs were found to influence ATS, while the NHIP was observed to affect AIW. In addition, a pathway via the NHIP affecting IEIBF was added, but this relationship was not significant. Individuals with environmental concerns have higher levels of sustainability awareness, as they are more likely to adopt eco-friendly behaviors and make conscious choices that minimize their environmental footprint [26]. Similarly, social norms have been shown to have a significant impact on sustainability. They play a role in guiding individuals and societies to adopt environmentally, economically and socially sustainable behaviors. Norms can encourage individuals to adopt behaviors such as recycling, saving energy, or consuming sustainable food [70,71].

Individuals who are more likely to integrate their needs into natural processes are also more likely to take animal welfare into account, as they tend to recognize the interconnectedness of human actions and the well-being of other living beings. Individuals who prioritize organic and sustainable food choices have been found to be more likely to avoid factory-farmed animal products because they associate these practices with ethical and ecological harm. This suggests that a deeper attunement with natural processes fosters greater sensitivity to the treatment of animals [72]. Similarly, while the NHIP does not directly influence the intention to consume insects, it has an indirect effect through ATS and AIW, suggesting that attitudes towards sustainability and animal welfare play a mediating role in shaping dietary choices.

By refining the model and removing weak links, a more robust structure was achieved. The strengthened relationship between AIW and IEIBF after controlling for age and gender suggests that demographic factors play a limited role in the model [43,44]. Although SNs do not directly influence the intention to consume EIBF, their significant effects on FNEO and NCIBF emphasize the role of social norms in shaping food preferences. Moreover, the observed negative impact of FNEO on NCIBF and the mitigating effect of social norms on food neophobia further highlight the complex interplay between consumer attitudes and external influences [46].

Overall, this study highlights the complex and multidimensional nature of individuals’ attitudes towards the consumption of insect-based foods (IBFs). Consumer preferences are influenced by a combination of personal attitudes, such as sustainability and animal welfare concerns, and external factors, such as social norms, cultural beliefs, and environmental awareness. While demographic factors such as age and gender play a minor role, social norms and food neophobia have a significant impact on individuals’ willingness to consume IBFs. The findings suggest that raising awareness of the environmental and health benefits of IBFs, as well as addressing psychological barriers and utilizing social norms, may help bridge the gap between intention and actual consumption. Further research is needed to validate these pathways and explore additional factors that may influence the uptake of IBFs in different contexts.

## 5. Conclusions

As a result of this study, we found that biospheric values are the cornerstone of the model; it is a strong variable that affects almost the whole chain. In particular, neophobia is a barrier that negatively affects consumption behavior and intention. Social norms can have a positive effect on neophobia and indirectly influence the attitude towards IBFs. The negative relationship between intention to consume and actual consumption (NCIBF) suggests that intention does not translate into behavior. This may be due to perceived risks (PRs) or other barriers. In conclusion, this model provides a very comprehensive analysis within the framework of an integrated sustainable neophilic IBF consumption model (ISNIEM).

To promote the consumption of IBFs as a sustainable alternative in Turkey, it is important to understand the attitudes and behaviors of society. Understanding consumers’ attitudes and behaviors toward IBFs and the factors that influence these attitudes and behaviors will play important roles in the production, marketing, and consumption of such foods in the future. In this study, we aimed to examine the factors influencing individuals’ intentions to consume IBFs using an integrated attitude–intention–eating behavior model. We observed positive and significant effects of sustainability attitudes on the intention to consume IBFs. Although sustainability awareness increases IEIBF, the misalignment between intention and actual preference suggests that Turkish society is not yet fully receptive to these foods, though this finding should be considered within this study’s limited generalizability. For individuals to accept such foods, more research is needed at both the production and marketing stages.

To increase the acceptance of IBFs in Turkey, practical strategies based on the findings of this study could be implemented. Targeted educational campaigns can help raise awareness about the sustainability and nutritional benefits of IBFs. These campaigns should specifically address FNEO by emphasizing the health and environmental benefits of IBFs, and by gradually introducing these foods in familiar contexts to reduce resistance. Additionally, culturally sensitive marketing approaches are essential to overcome societal and religious barriers. Tailored messaging that aligns with the values and beliefs of Turkish society can enhance the appeal of IBFs. Collaboration with local influencers and food industry stakeholders can further facilitate the acceptance process by promoting these foods in a way that respects cultural norms. Overall, a comprehensive strategy that combines education, social influence, and culturally relevant marketing will likely contribute to increasing IBF consumption in Turkiye.

## 6. Limitations

This study has several limitations that should be acknowledged. The use of an online survey restricted participation to individuals with internet access, potentially underrepresenting rural populations or those with limited digital engagement. This may have skewed findings, as attitudes toward insect-based foods (IBFs) could vary across socio-economic and geographical segments. The sample was also predominantly highly educated and female, limiting generalizability to a broader population. Future research could enhance representativeness by conducting field studies and employing stratified and cluster sampling techniques to ensure a more balanced and inclusive sample.

While this study integrates key socio-psychological and sustainability-related factors, it does not fully explore the influence of cultural and religious beliefs on IBF consumption. Given their strong role in shaping dietary choices, multidisciplinary studies should examine these aspects in greater depth. Additionally, the reliance on an analytical design rather than qualitative methods limits the ability to uncover deeper motivations, perceptions, and barriers to IBF consumption. This study also reveals a disconnect between intention and actual behavior, suggesting psychological or contextual factors that prevent translation into action. Understanding these barriers through behavioral interventions or experimental research could provide valuable insights. Furthermore, as a cross-sectional study, this research captures attitudes at a single point in time, making it difficult to establish causality. Longitudinal studies are needed to track how exposure, knowledge, and social norms influence IBF consumption over time and assess the long-term impact of targeted educational campaigns and policy interventions.

Despite these limitations, we believe this study provides significant evidence on the socio-psychological and sustainability-related factors influencing IBF consumption. By integrating a comprehensive theoretical model and testing it in a country where IBFs are not yet widely accepted, this research offers valuable insights that can inform future policies, marketing strategies, and consumer education efforts to promote sustainable dietary shifts.

## Figures and Tables

**Figure 1 foods-14-00984-f001:**
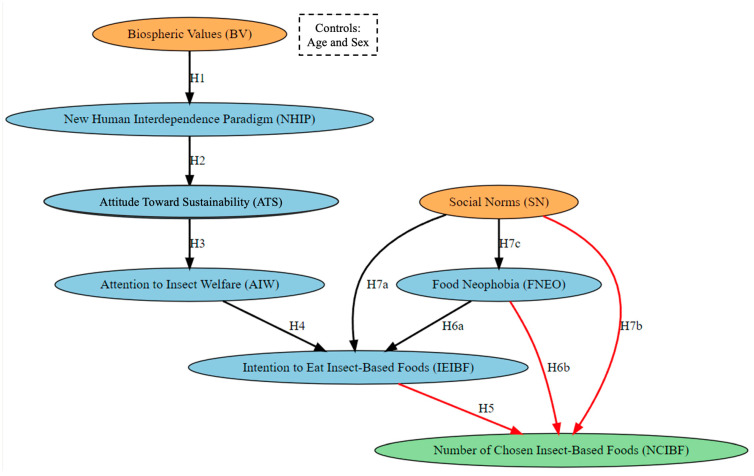
Theoretical model.

**Figure 2 foods-14-00984-f002:**
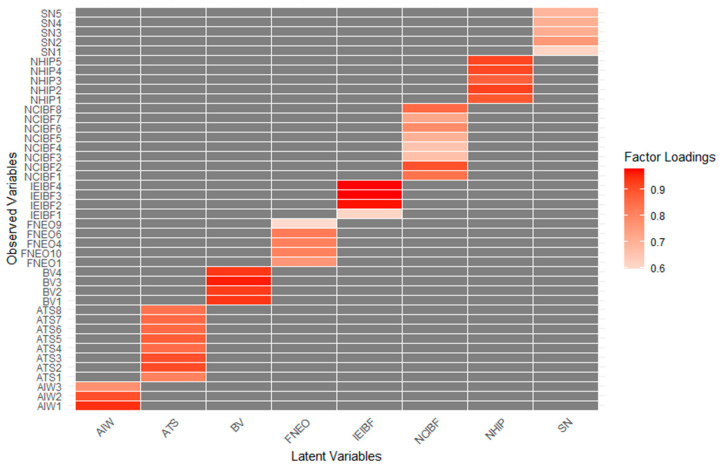
Factor loadings and defined covariances in CFA prior to SEM model.

**Figure 3 foods-14-00984-f003:**
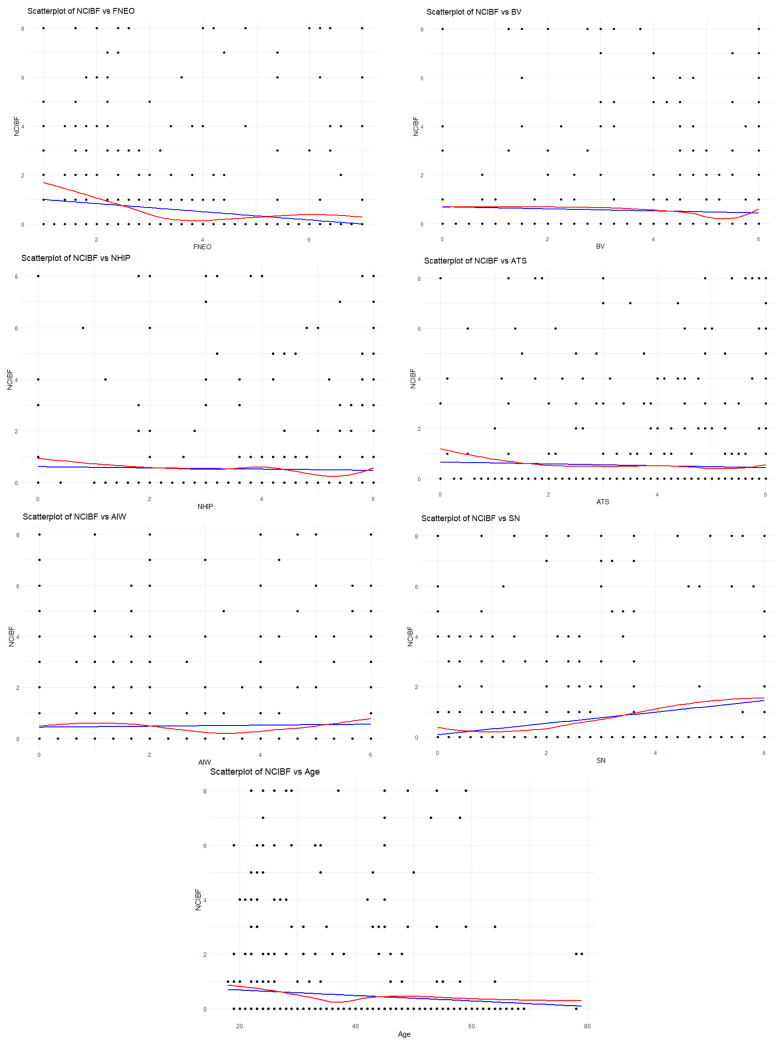
Scatter plot of NCIBF and independent variables.

**Figure 4 foods-14-00984-f004:**
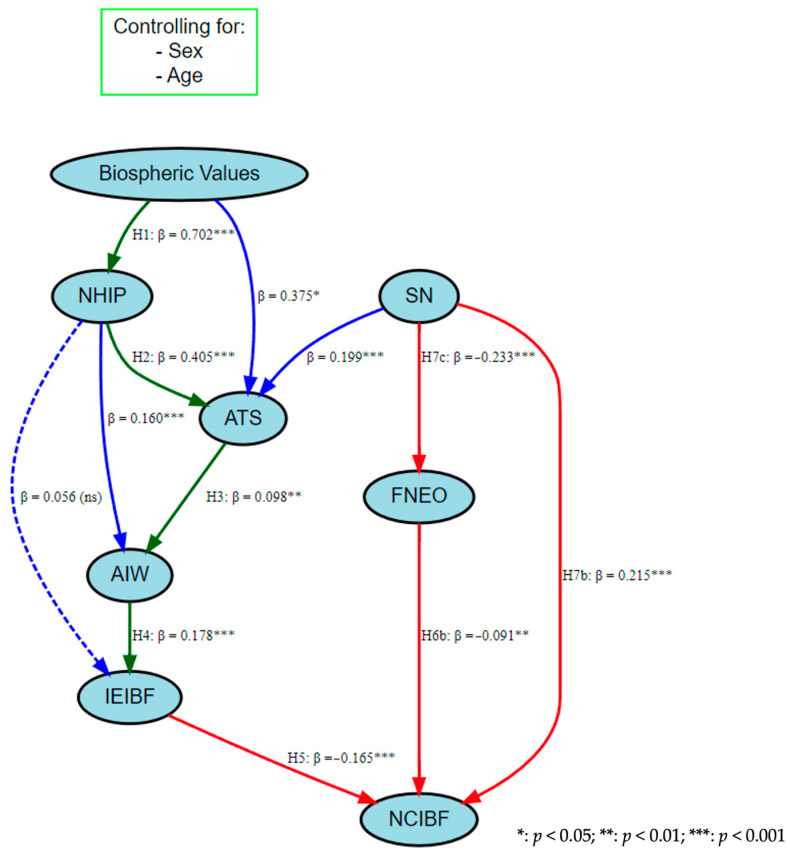
Final integrated sustainable neophobic insect-based eating model.

**Table 1 foods-14-00984-t001:** Items and definitions of the variables evaluated in this study.

Variable	Definition	Items
Biospheric Values	Questions the value placed on the protection of the environment and the biosphere.	BV1: Protecting nature (protecting the environment) BV2: Feeling part of the natural environment BV3: Protecting natural resources (preventing pollutions) BV4: Being compatible with other species (respectful to the World)
New Human Interdependence Paradigm	Includes the interdependence between human progress and conservation, conceived as a dynamic process of integrating human needs into natural processes.	NHIP1: Real human progress can only be achieved by maintaining the ecological balance. NHIP2: Protecting nature today means securing the future of humanity. NHIP3: We must reduce our consumption level to ensure the prosperity of current and future generations. NHIP4: Humanity can only progress by protecting natural resources. NHIP5: People can only enjoy nature if they use their resources wisely.
Attitude Toward Sustainability	Draws attention to sustainability themes by combining the environmental impact indices of the food chain with indicators of social and economic sustainability.	ATS1: Organic production ATS2: Use of alternative energy sources ATS3: Being recycled packaged ATS4: Carbon footprint hardening (low carbon emission) ATS5: Water footprint certification (limited water use) ATS6: Being in a short supply chain ATS7: Being of local (local) origin ATS8: Low use of chemical compounds (such as pesticides)
Social Norms	Questions various norms, including descriptive (what significant others do) and injunctive (beliefs about what is officially prescribed and the expectations of very important others).	SN1: Most of my loved ones would approve of me trying alternative proteins to meat derived from insect meat. SN2: Most of my loved ones reduced their meat consumption. SN3: Most of my friends would approve of my preference for eating insect protein by reducing meat consumption. SN4: No matter what others do, I feel morally obliged to reduce my consumption of meat or meat. SN5: I feel good if I avoid eating meat during a meal.
IEIBF-1	Includes the IEIBF and the acceptable conditions associated with this choice.	IEIBF1: I never eat insect products because I find them repulsive. IEIBF2: If this becomes a norm in the city where I live, I will eat insect products. IEIBF3: I only eat insect products if they are cooked by prestigious chefs. IEIBF4: I would only like to eat processed insect products (for example, cookies, snacks, hamburgers, etc.) in which the insect does not appear.
Intention to Eat Insect-Based Food-2		IEIBF1: I would consider buying insect-based products because their protein value is higher. EIBF2: I would consider switching to insect-based products for ecological reasons. EIBF3: I can spend more money on insect-based products than on traditional products. EIBF4: I hope to buy insect-based products in the future because they have a positive effect on the environment. EIBF5: I definitely plan to buy insect-based products in the near future.
Perceived Risks	Concerns about the consumption of insect-based foods are questioned from a health, economic, and expectations perspective.	PR1: I am worried that if I consume insect-based products, it will harm or hurt my body. PR2: I am worried that insect-based products will not be of good value for my money. PR3: When I use insect-based products, I worry about hygiene/health issues. PR4: I worry about being disappointed because of my experience of using insect-based products.
Attention to Insect Welfare	Focuses on ethical values and welfare in the life cycle of insects.	AIW1: It is important to maintain the well-being of insects during breeding work (such as spraying). AIW2: The well-being of the insect is less important than that of other farm animals (cattle, pig, poultry, small cattle, fish). AIW3: Ethically carried out insecticultivation can be pursued up to higher product quality.
Number of selected IBFs	Foods are presented visually and participants are asked which ones they can consume. The aim was to assess the acceptability of insect-based foods using various alternatives, to choose between ready-to-eat products available on the market and products to be prepared, and to create a basket of alternatives that consumers can utilize for consumption. The main objective is to assess the actual acceptability of different forms of the product.	NCIBF1: Insect meatball NCIBF2: Insect burger NCIBF3: Bread with grains NCIBF4: Spaghetti with crickets NCIBF5: Fried grasshoppers NCIBF6: Muffin with cricket flour NCIBF7: Fried insects NCIBF8: Crackers with cricket flour
Neophobia	Evaluates the attitude of individuals toward unfamiliar foods.	N1. I always try new and different foods. N2. I do not trust new foods. N3. I do not try a food that I do not know what is in it. N4. I like the foods of different countries. N5. It feels very strange to consume foods from different countries N6. I try new food for dinner invitations. N7. I am afraid of consuming foods I have never consumed before. N8. I am very picky about the foods I will consume. N9. I eat almost everything. N10. I like to try restaurants from different countries.
New Human Interdependence Paradigm	Includes the interdependence between human progress and conservation, conceived as a dynamic process of integrating human needs into natural processes.	NHIP1: Real human progress can only be achieved by maintaining the ecological balance. NHIP2: Protecting nature today means securing the future of humanity. NHIP3: We must reduce our consumption level to ensure the prosperity of current and future generations. NHIP4: Humanity can only progress by protecting natural resources. NHIP5: People can only enjoy nature if they use their resources wisely.
Attitude Toward Sustainability	Draws attention to sustainability themes by combining the environmental impact indices of the food chain with indicators of social and economic sustainability.	ATS1: Organic production ATS2: Use of alternative energy sources ATS3: Being recycled packaged ATS4: Carbon footprint hardening (low carbon emission) ATS5: Water footprint certification (limited water use) ATS6: Being in a short supply chain ATS7: Being of local (local) origin ATS8: Low use of chemical compounds (such as pesticides)
Social Norms	Questions various norms, including descriptive (what significant others do) and injunctive (beliefs about what is officially prescribed and the expectations of very important others).	SN1: Most of my loved ones would approve of me trying alternative proteins to meat derived from insect meat. SN2: Most of my loved ones reduced their meat consumption. SN3: Most of my friends would approve of my preference for eating insect protein by reducing meat consumption. SN4: No matter what others do, I feel morally obliged to reduce my consumption of meat or meat. SN5: I feel good if I avoid eating meat during a meal.
IEIBF-1	Includes the IEIBF and the acceptable conditions associated with this choice.	IEIBF1: I never eat insect products because I find them repulsive. IEIBF2: If this becomes a norm in the city where I live, I will eat insect products. IEIBF3: I only eat insect products if they are cooked by prestigious chefs. IEIBF4: I would only like to eat processed insect products (for example, cookies, snacks, hamburgers, etc.) in which the insect does not appear.
Intention to Eat Insect-Based Food-2		IEIBF1: I would consider buying insect-based products because their protein value is higher. EIBF2: I would consider switching to insect-based products for ecological reasons. EIBF3: I can spend more money on insect-based products than on traditional products. EIBF4: I hope to buy insect-based products in the future because they have a positive effect on the environment. EIBF5: I definitely plan to buy insect-based products in the near future.
Perceived Risks	Concerns about the consumption of insect-based foods are questioned from a health, economic, and expectations perspective.	PR1: I am worried that if I consume insect-based products, it will harm or hurt my body. PR2: I am worried that insect-based products will not be of good value for my money. PR3: When I use insect-based products, I worry about hygiene/health issues. PR4: I worry about being disappointed because of my experience of using insect-based products.
Attention to Insect Welfare	Focuses on ethical values and welfare in the life cycle of insects.	AIW1: It is important to maintain the well-being of insects during breeding work (such as spraying). AIW2: The well-being of the insect is less important than that of other farm animals (cattle, pig, poultry, small cattle, fish). AIW3: Ethically carried out insecticultivation can be pursued up to higher product quality.
Number of selected IBFs	Foods are presented visually and participants are asked which ones they can consume. The aim was to assess the acceptability of insect-based foods using various alternatives, to choose between ready-to-eat products available on the market and products to be prepared, and to create a basket of alternatives that consumers can utilize for consumption. The main objective is to assess the actual acceptability of different forms of the product.	NCIBF1: Insect meatball NCIBF2: Insect burger NCIBF3: Bread with grains NCIBF4: Spaghetti with crickets NCIBF5: Fried grasshoppers NCIBF6: Muffin with cricket flour NCIBF7: Fried insects NCIBF8: Crackers with cricket flour
Neophobia	Evaluates the attitude of individuals toward unfamiliar foods.	N1. I always try new and different foods. N2. I do not trust new foods. N3. I do not try a food that I do not know what is in it. N4. I like the foods of different countries. N5. It feels very strange to consume foods from different countries N6. I try new food for dinner invitations. N7. I am afraid of consuming foods I have never consumed before. N8. I am very picky about the foods I will consume. N9. I eat almost everything. N10. I like to try restaurants from different countries.

**Table 2 foods-14-00984-t002:** Participant characteristics.

			n	%	
**Sex**					
Female			824	68.9	
Male			356	29.77	
I do not want to specify			16	1.34	
**Paying attention to the monthly purchase of green products**					
None/very little			74	6.19	
Less			188	15.72	
A little bit			486	40.64	
More			349	29.18	
Too much			99	8.28	
**Sustainability of purchased products**					
None/very little			89	7.44	
Less			140	11.71	
A little bit			523	43.73	
More			339	28.34	
Too much			105	8.78	
**Purchased products are healthy**					
None/very little			16	1.34	
Less			45	3.76	
A little bit			221	18.48	
More			591	49.41	
Too much			323	27.01	
**Education level**					
Primary–secondary school			34	2.84	
Secondary school graduate			197	16.47	
Associate degree			120	10.03	
Bachelor’s degree			590	49.33	
Postgraduate (master’s/doctorate/specialization)			255	21.32	
**Marital status**					
Married			695	58.11	
Single			501	41.89	
**Occupation**					
Civil servant			323	27.01	
Student			197	16.47	
Employer			70	5.85	
Private sector employee			299	25	
Self-employed			87	7.27	
Unemployed/not working			35	2.93	
Housewife			97	8.11	
Retiree			88	7.36	
**Monthly Income**					
TRY 13,400 TL–20,000			182	15.22	
TRY 20,000–30,000			268	22.41	
TRY 30,000–40,000			188	15.72	
TRY 40,000 vand more			252	21.07	
I am a student and receive pocket money from my parents			147	12.29	
Around the minimum wage			113	9.45	
Below the minimum wage			46	3.85	
**Previous consumption of insect-containing products**					
No			1074	89.95	
Yes			120	10.05	
*Total*			1194	100	
	**n**	**Min**	**Max**	**Mean**	**Standard Deviation**
**Age**	1196	18	79	38.4	12.48

**Table 3 foods-14-00984-t003:** Summary of exploratory factor analysis results.

Factor	SS Loadings	Explained Variance	KMO	Bartlett’s Chi-Square	Factor Loadings of Items
BVs	3.53	0.88	0.87	6176.2; *p* < 0.001	0.94|0.93|0.96|0.93
NHIP	4.06	0.81	0.9	6485.18; *p* < 0.001	0.88|0.92|0.88|0.91|0.92
ATS	5.96	0.74	0.94	9980.69; *p* < 0.001	0.79|0.91|0.9|0.87|0.9|0.86|0.85|0.82
AIW	2.29	0.76	0.72	2476.13; *p* < 0.001	0.95|0.9|0.77
SNs	2.63	0.53	0.78	2661.43; *p* < 0.001	0.67|0.7|0.76|0.75|0.74
IEIBF	3.2	0.8	0.83	6281.59; *p* < 0.001	0.61|0.97|0.98|0.97
FNEO	2.91	0.58	0.85	2908.78; *p* < 0.001	0.76|0.81|0.82|0.6|0.81
NCIBF	4.91	0.61	0.85	8428.24; *p* < 0.001	0.89|0.92|0.69|0.68|0.7|0.78|0.71|0.84

**Table 4 foods-14-00984-t004:** Convergent validity and reliability scores.

Construct	Items	CFA Loadings	ωH	FLC	CR	AVE	Alpha	Μ ± Σ Min–Max
Biospheric Values (BVs)	BV1	0.937	0.97	0.939 > 0751 (NHIP)	0.968	0.882	0.968	4.54 ± 0.37 0–6
	BV2	0.928						
	BV3	0.961						
	BV4	0.930						
New Human Interdependence Paradigm (NHIP)	NHIP1	0.879	0.96	0.900 > 0751 (BV)	0.956	0.812	0.955	4.85 ± 0.41 0–6
	NHIP2	0.915						
	NHIP3	0.876						
	NHIP4	0.914						
	NHIP5	0.921						
Attitude Toward Sustainability (ATS)	ATS1	0.788	0.96	0.864 > 0.686 (NHIP)	0.959	0.745	0.958	4.28 ± 0.71 0–6
	ATS2	0.907						
	ATS3	0.903						
	ATS4	0.873						
	ATS5	0.898						
	ATS6	0.858						
	ATS7	0.846						
	ATS8	0.824						
Attention to Insect Welfare (AIW)	AIW1	0.906		0.874 > 0.221 (IEIBF)	0.948	0.896	0.769	2.89 ± 0.52 0–6
	AIW2	0.764	0.91					
	AIW3	0.903						
Neophobia (FNEO)	FNEO1	0.757	0.87	0.764 > 0.265 (SN)	0.873	0.582	0.870	4.03 ± 1.21 1–7
	FNEO2 (Removed)	-						
	FNEO3 (Removed)	-						
	FNEO4	0.807						
	FNEO5 (Removed)	-						
	FNEO6	0.820						
	FNEO7 (Removed)	-						
	FNEO8 (Removed)	-						
	FNEO9	0.595						
	FNEO10	0.812						
Social Norms (SNs)	SN1	0.672	0.85	0.712 > 0.374 (ATS)	0.847	0.526	0.845	1.80 ± 1.13 0–6
	SN2	0.700						
	SN3	0.762						
	SN4	0.746						
	SN5	0.742						
Intention to Eat Insects Based Foods (IEIBF)	IEIBF1	0.605	0.94	0.905 > 0.221 (SN)	0.940	0.801	0.931	3.16 ± 0.61 0–6
	IEIBF2	0.966						
	IEIBF3	0.977						
	IEIBF4	0.975						
Number of Chosen IBFs (NCIBF)	NCIBF1	0.890	0.93	0.776 > 0.217 (SN)	0.926	0.614	0.925 (KR20:0.929)	0.50 ± 1.56 0–8
	NCIBF2	0.925						
	NCIBF3	0.692						
	NCIBF4	0.684						
	NCIBF5	0.700						
	NCIBF6	0.782						
	NCIBF7	0.714						
	NCIBF8	0.843						

CR: coefficient of reliability; AVE: Average Variance Extracted; FLC: Fornell–Larcker criterion.

**Table 5 foods-14-00984-t005:** Bivariate correlation matrix of the factor variables.

	NHIP		ATS		AIW		SN		IEIBF		FNEO		NCIBF	
	r	*p*	R	*p*	r	*p*	r	*p*	R	*p*	r	*p*	r	*p*
BV	**0.668 _sp_**	**<0.001 ****	**0.599 _sp_**	**<0.001 ****	**0.167 _sp_**	**<0.001 ****	**0.196 _sp_**	**<0.001 ****	**0.190 _sp_**	**<0.001 ****	**−0.143 _sp_**	**<0.001 ****	−0.033 _sp_	0.253
NHIP	1		**0.641 _sp_**	**<0.001 ****	**0.182 _sp_**	**<0.001 ****	**0.233 _sp_**	**<0.001 ****	**0.144 _sp_**	**<0.001 ****	**−0.204 _sp_**	**<0.001 ****	−0.003 _sp_	0.915
ATS			1		**0.177 _sp_**	**<0.001 ****	**0.329 _pe_**	**<0.001 ****	**0.169 _sp_**	**<0.001 ****	**−0.232 _pe_**	**<0.001 ****	−0.053 _sp_	0.065
AIW					1		0.052 _sp_	0.071	**0.212 _sp_**	**<0.001 ****	**−0.132 _sp_**	**<0.001 ****	−0.015 _sp_	0.610
SN							1		0.052 _sp_	0.071	**−0.238 _pe_**	**<0.001 ****	**0.180 _sp_**	**<0.001 ****
IEIBF									1		**−0.070 _sp_**	**0.015 ***	**−0.159 _sp_**	**<0.001 ****
FNEO											1		**−0.237 _sp_**	**<0.001 ****
NCIBF													1	

sp, Spearman’s rho; pe, Pearson’s r; * *p* < 0.05; ** *p* < 0.001, BV, biospheric values; NHIP, new human interdependence paradigm; ATS, attitude toward sustainability; AIW, attention to insect welfare; SNs, social norms; IEIBF, intention to eat insect-based foods; FNEO, food neophobia; NCIBF: number of chosen insect-based foods.

**Table 6 foods-14-00984-t006:** CFA fit indices and common method bias assessment prior to structural equation modeling.

Fit Index and Thresholds Used	Analysis Value
χ^2^/df ≤ 5.00	4.07
0.90 ≤ CFI ≤ 1.00	0.936
0.90 ≤ Robust CFI ≤ 1.00	0.939
0.90 ≤ TLI ≤ 1.00	0.930
0.90 ≤ Robust TLI ≤ 1.00	0.933
RMSEA < 0.08	0.051
Robust RMSEA < 0.08	0.056
sRMR < 0.08	0.055
0.85 ≤ GFI ≤ 1.00	0.922
0.85 ≤ AGFI ≤ 1.00	0.914
**TVE for Common Bias < 0.50**	**0.28**

**Table 7 foods-14-00984-t007:** Variance Inflation Factor results.

Variable	VIF Value
BV	2.290271
NHIP	2.518469
ATS	2.104584
AIW	1.087773
IEIBF	1.095272
FNEO	1.234429
SN	1.171359
Age	1.131385

**Table 8 foods-14-00984-t008:** Determination of linear and non-linear relationships.

Variable	Beta	SE	t	*p*
Intercept	1.060451	0.164186	6.459	<0.001 ***
BV (poly 1)	−1.39768	2.22031	−0.629	0.52915
BV (poly 2)	0.908774	1.839957	0.494	0.62146
NHIP (poly 1)	2.931351	2.344714	1.25	0.21148
NHIP (poly 2)	1.311345	1.967864	0.666	0.5053
ATS (poly 1)	−6.06845	2.147909	−2.825	0.0048 **
ATS (poly 2)	−0.88159	1.79202	−0.492	0.62285
AIW (poly 1)	0.425247	1.544671	0.275	0.78314
AIW (poly 2)	1.345137	1.52451	0.882	0.37777
FNEO (poly 1)	−6.89477	1.65014	−4.178	<0.001 ***
FNEO (poly 2)	5.162567	1.595761	3.235	0.00125 **
SN (poly 1)	12.83651	1.631621	7.867	<0.001 ***
SN (poly 2)	3.032774	1.565905	1.937	0.05302
IEIBF (poly 1)	−7.72157	1.549741	−4.982	<0.001 ***
IEIBF (poly 2)	0.417488	1.552511	0.269	0.78805
Age	−0.00707	0.003647	−1.938	0.0529
Sex	−0.41715	0.096478	−4.324	<0.001 ***

Polynomial and linear mixed model. Dependent: NCIBF; **: *p* < 0.01; ***: *p* < 0.001.

**Table 9 foods-14-00984-t009:** SEM results: theoretical model, modifications, and covariances.

	Variable	Beta	Std Error	Std Beta	z	*p*	Dependent Variable
Theoretical Model	BV	0.625	0.027	0.702	22.791	<0.001 ***	NHIP
	Age _R_	−0.000	0.003	−0.002	−0.062	0.951	NHIP
	Sex (0:Male)	0.169	0.064	0.058	2.251	0.008 **	NHIP
	NHIP	0.439	0.042	0.405	10.470	<0.001 ***	ATS_poly1
	Age	0.012	0.003	0.094	4.356	<0.001 ***	ATS_poly1
	Sex (0:Male)	0.179	0.075	0.057	2.388	0.017 *	ATS_poly1
	ATS_poly1st	0.129	0.049	0.098	2.604	0.009 **	AIW
	Age	−0.009	0.005	−0.122	−1.955	0.051	AIW
	Sex (0:Male)	−0.505	0.115	−0.056	−4.395	<0.001 ***	AIW
	SN_poly1st	−2.099	0.236	−0.233	−8.898	<0.001 ***	FNEO_poly2nd
	Age	0.294	0.032	0.262	9.192	<0.001 ***	FNEO_poly2nd
	Sex (0:Male)	−1.989	0.808	−0.071	−1.989	0.014 *	FNEO_poly2nd
	SN_poly1st _R_	0.028	0.046	0.019	0.613	0.540	IEIBF_poly1
	FNEO_poly2nd _R_	0.002	0.005	0.011	0.358	0.720	IEIBF_poly1
	AIW	0.199	0.033	0.178	6.121	<0.001 ***	IEIBF_poly1
	Age _R_	−0.008	0.005	−0.041	−1.434	0.151	IEIBF_poly1
	Sex (0:Male)	0.204	0.137	0.044	1.487	0.137	IEIBF_poly1
	IEIBF_poly1st	−0.112	0.018	−0.165	−6.133	<0.001 ***	NCIBF
	FNEO_poly2nd	−0.010	0.003	−0.091	−2.943	0.003 **	NCIBF
	SN_poly1st	0.216	0.039	0.215	5.567	<0.001 ***	NCIBF
	Age	−0.010	0.004	−0.083	−2.933	0.003 **	NCIBF
	Sex (0:Male)	−0.426	0.107	−0.135	−3.982	<0.001 ***	NCIBF
Modifications	BV	0.365	0.170	0.375	2.153	0.031 *	ATS_poly1
	SN_poly1st	0.203	0.028	0.199	7.379	<0.001 ***	ATS_poly1
	NHIP	0.224	0.051	0.160	4.410	<0.001 ***	AIW
	NHIP	0.088	0.076	0.056	1.151	0.250	IEIBF_poly1
	Covariances:						
	FNEO_poly2	−1.952	0.447	−0.147	−4.371	<0.001 ***	cov(NHIP)
	IEIBF_poly1	−0.284	0.101	−0.127	−2.804	0.005 **	cov(NHIP)

*: *p* < 0.05; **: *p* < 0.01; ***: *p* < 0.001; _R_: removed from model path.

**Table 10 foods-14-00984-t010:** SEM model: Goodness-of-Fit Indices.

Fit Index and Thresholds Used	Analysis Value
χ^2^/df ≤ 5.00	3.06
0.90 ≤ CFI ≤ 1.00	0.967
0.90 ≤ Robust CFI ≤ 1.00	0.968
0.90 ≤ TLI ≤ 1.00	0.914
0.90 ≤ Robust TLI ≤ 1.00	0.916
RMSEA < 0.08	0.066
Robust RMSEA < 0.08	0.065
sRMR < 0.08	0.045
0.85 ≤ GFI ≤ 1.00	0.961
0.85 ≤ AGFI ≤ 1.00	0.985

## Data Availability

Data are contained within the article.

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
