# Peer review of "Factors Influencing the Intention to Eat Insects as an Alternative Protein Source: A Sample from Turkey"

_foods, 2025, doi:10.3390/foods14060984_

Round 1
Reviewer 1 Report
Comments and Suggestions for Authors
The study investigate in a comprehensive manner the factors influencing the intention to eat insect based food in a Turkey sample of participants. The article is well structured, the research protocol is rigorous established and includes detalied statistical analysis, the findings are relevant and follow the hypotheses tested in this study.
However, several sections require attention to enhance the clarity of presented data and accomplish the accuracy and scientific rigor:
There are many abbreviations in the manuscript, explained when they first appear in the text, this can be difficult to follow and sometimes confusing. By example, the abbreviation for "insect-based foods" is explained only on page 8, altough IBF appears for the first time in a table on page 6, which may raise questions for readers. To avoid these inconveniences, please also provide the list of abbreviations on the first page, immediately after the abstract.
In the Conclusion section, it's not clearly explained which factors influence the intention to consume insects among the subjects of the present study. Only sustainability attitudes are mentiones, without any explanation. Please complete the section accordingly this suggestion (include the summary of research findings).
Author Response
Thank you for your valuable feedback. We acknowledge that the revisions you suggested will contribute significantly to our manuscript, and you can be assured that we aim to implement them in the best possible way. If there are any remaining shortcomings, please inform us, and we will make further adjustments. With this in mind, we respectfully submit the following revisions we have made:
Comment 1: There are many abbreviations in the manuscript, explained when they first appear in the text, this can be difficult to follow and sometimes confusing. By example, the abbreviation for "insect-based foods" is explained only on page 8, altough IBF appears for the first time in a table on page 6, which may raise questions for readers. To avoid these inconveniences, please also provide the list of abbreviations on the first page, immediately after the abstract.
Response 1: We had the same hesitation about abbreviations, but we had to give the terms together with their abbreviations because we thought that the repetition of similar words would tire the reader. Since the abbreviations are at the bottom of the template, we presented them there, if there is no problem for the publication, of course, we can give it above. (which we have done as you requested)
Comment 2: In the Conclusion section, it's not clearly explained which factors influence the intention to consume insects among the subjects of the present study. Only sustainability attitudes are mentiones, without any explanation. Please complete the section accordingly this suggestion (include the summary of research findings).
Response 2: We have made the arrangements as per your request. You can find it on lines 504-512.
Reviewer 2 Report
Comments and Suggestions for Authors
Dear Authors,
The introduction of insects in human’s alimentation is a current subject and its in-depth research is relevant. I appreciate the clearness of the ideas included in the manuscript.
Though, there are some elements that need a more thorough analysis.
1. At lines 82-86 you mentioned that no studies have been organized until now to address attitudinal and behavioral intentions to eat insects. The cited article was published in 2017, 8 years ago, a large time interval in which many other articles may have been published on this topic. That is why the assumption needs more recent elements to be trustworthy and correct.
2. The Literature analysis lacks in consistency and in explaining the variables integrated in the study. All concepts included in the hypotheses need to be theoretically present prior their effective assessment. At the same time, the correlations between the variables which were included into hypotheses, have to be explained based on the literature review.
3. You mentioned that you have implemented the questionnaire of an Italian study – reference 22, without detailing its importance and without explaining why you decided to use it. Moreover, aren’t there any differences for attitudes and behavior between Italian and Turkish consumers to refine the questions?
4. At line 153 there is a mention for R 4.3.1. which I didn’t find in the manuscript
5. Discussion section is a weak part of the manuscript. The effective results are vaguely analyzed, instead there are included unimportant parallels to other studies.
6. Limitations of the study and further needed studies on the topic have to be included in the manuscript.
Author Response
Thank you for your valuable feedback. We acknowledge that the revisions you suggested will contribute significantly to our manuscript, and you can be assured that we aim to implement them in the best possible way. If there are any remaining shortcomings, please inform us, and we will make further adjustments. With this in mind, we respectfully submit the following revisions we have made:
Comment 1: At lines 82-86 you mentioned that no studies have been organized until now to address attitudinal and behavioral intentions to eat insects. The cited article was published in 2017, 8 years ago, a large time interval in which many other articles may have been published on this topic. That is why the assumption needs more recent elements to be trustworthy and correct.
Response 1: We wanted to state that there are studies in this field in the literature, but a comprehensive study has not yet been carried out in Turkey. We have edited the sentence, you can find it in line 94.
Comment 2 : The Literature analysis lacks in consistency and in explaining the variables integrated in the study. All concepts included in the hypotheses need to be theoretically present prior their effective assessment. At the same time, the correlations between the variables which were included into hypotheses, have to be explained based on the literature review.
Response 2: The concepts in the hypotheses are defined and detailed in both the “data collection procedure” and “table 1.” Correlations between the variables are both included in the introduction (lines 99-108) and compared with our findings in the discussion section.
Comment 3: You mentioned that you have implemented the questionnaire of an Italian study – reference 22, without detailing its importance and without explaining why you decided to use it. Moreover, aren’t there any differences for attitudes and behavior between Italian and Turkish consumers to refine the questions?
Response 3: Of course there are differences. However, cultural adaptation was ensured during translation. In addition, these questions were chosen because they are the most comprehensive questionnaire developed in this field and because the food cultures of Italians and Turks are close to each other, albeit with minor differences. (lines 173-176)
Comment 4: At line 153 there is a mention for R 4.3.1. which I didn’t find in the manuscript.
Response 4: Thank you for pointing this out. The mention of R 4.3.1 is included in the manuscript, specifically where we describe the packages used for data analysis (lavaan, psych, semTools, ggplot2, GGally, diagrammer). We have also indicated that coding was performed and included additional packages for visualization. We have reviewed the manuscript to ensure that this information is correctly referenced and clear. If there is still any ambiguity, we would be happy to provide further details. To build trust, we are also sharing the codes with you:
library(readxl)
library(psych)
library(lavaan)
rm(list = ls())
data <- read_excel("C:/Users/Salim/OneDrive/Desktop/istatistik isleri/Murat Bas bocek tuketimi/veriler.xlsx")
calculate_CR_AVE <- function(loadings, error_variances) {
CR <- (sum(loadings)^2) / ((sum(loadings)^2) + sum(error_variances))
AVE <- mean(loadings^2)
return(list(CR = CR, AVE = AVE))
}
# biospheric
biospheric_values_model <- 'BiosphericValues =~ BiosphericValues1 + BiosphericValues2 + BiosphericValues3 + BiosphericValues4'
biospheric_fit <- cfa(biospheric_values_model, data = data)
biospheric_loadings <- lavInspect(biospheric_fit, "std")$lambda
biospheric_loadings <- as.numeric(biospheric_loadings)
biospheric_error_variances <- lavInspect(biospheric_fit, "std")$theta
biospheric_error_variances <- as.numeric(diag(biospheric_error_variances))
biospheric_results <- calculate_CR_AVE(biospheric_loadings, biospheric_error_variances)
biospheric_alpha <- psych::alpha(data[, c("BiosphericValues1", "BiosphericValues2", "BiosphericValues3", "BiosphericValues4")])$total$raw_alpha
biospheric_results$CR
biospheric_results$AVE
biospheric_alpha
biospheric_standardized_loadings <- lavInspect(biospheric_fit, "std")$lambda
biospheric_standardized_loadings
biospheric_values_mean <- rowMeans(data[, c("BiosphericValues1", "BiosphericValues2", "BiosphericValues3", "BiosphericValues4")])
biospheric_values_sd <- apply(data[, c("BiosphericValues1", "BiosphericValues2", "BiosphericValues3", "BiosphericValues4")], 1, sd)
biospheric_summary <- paste0("M????: ", round(mean(biospheric_values_mean), 2), " + ", round(mean(biospheric_values_sd), 2),
", Min-Max: ", round(min(biospheric_values_mean), 2), "-", round(max(biospheric_values_mean), 2))
biospheric_summary
# nhip
nhip_model <- 'NHIP =~ NewHumanInterdependenceParadigm1 + NewHumanInterdependenceParadigm2 + NewHumanInterdependenceParadigm3 + NewHumanInterdependenceParadigm4 + NewHumanInterdependenceParadigm5'
nhip_fit <- cfa(nhip_model, data = data)
nhip_loadings <- lavInspect(nhip_fit, "std")$lambda
nhip_loadings <- as.numeric(nhip_loadings)
nhip_error_variances <- lavInspect(nhip_fit, "std")$theta
nhip_error_variances <- as.numeric(diag(nhip_error_variances))
nhip_results <- calculate_CR_AVE(nhip_loadings, nhip_error_variances)
nhip_alpha <- alpha(data[, c("NewHumanInterdependenceParadigm1", "NewHumanInterdependenceParadigm2", "NewHumanInterdependenceParadigm3", "NewHumanInterdependenceParadigm4", "NewHumanInterdependenceParadigm5")])$total$raw_alpha
nhip_results$CR
nhip_results$AVE
nhip_alpha
nhip_standardized_loadings <- lavInspect(nhip_fit, "std")$lambda
nhip_standardized_loadings
nhip_mean <- rowMeans(data[, c("NewHumanInterdependenceParadigm1", "NewHumanInterdependenceParadigm2", "NewHumanInterdependenceParadigm3", "NewHumanInterdependenceParadigm4", "NewHumanInterdependenceParadigm5")])
nhip_sd <- apply(data[, c("NewHumanInterdependenceParadigm1", "NewHumanInterdependenceParadigm2", "NewHumanInterdependenceParadigm3", "NewHumanInterdependenceParadigm4", "NewHumanInterdependenceParadigm5")], 1, sd)
nhip_summary <- paste0("M????: ", round(mean(nhip_mean), 2), " + ", round(mean(nhip_sd), 2),
", Min-Max: ", round(min(nhip_mean), 2), "-", round(max(nhip_mean), 2))
nhip_summary
# attitude toward sustainability
sustainability_model <- 'Sustainability =~ AttitudeTowardSustainability1 + AttitudeTowardSustainability2 + AttitudeTowardSustainability3 + AttitudeTowardSustainability4 + AttitudeTowardSustainability5 + AttitudeTowardSustainability6 + AttitudeTowardSustainability7 + AttitudeTowardSustainability8'
sustainability_fit <- cfa(sustainability_model, data = data)
sustainability_loadings <- lavInspect(sustainability_fit, "std")$lambda
sustainability_loadings <- as.numeric(sustainability_loadings)
sustainability_error_variances <- lavInspect(sustainability_fit, "std")$theta
sustainability_error_variances <- as.numeric(diag(sustainability_error_variances))
sustainability_results <- calculate_CR_AVE(sustainability_loadings, sustainability_error_variances)
sustainability_alpha <- alpha(data[, c("AttitudeTowardSustainability1", "AttitudeTowardSustainability2", "AttitudeTowardSustainability3", "AttitudeTowardSustainability4", "AttitudeTowardSustainability5", "AttitudeTowardSustainability6", "AttitudeTowardSustainability7", "AttitudeTowardSustainability8")])$total$raw_alpha
summary(sustainability_fit)
sustainability_results$CR
sustainability_results$AVE
sustainability_alpha
sustainability_standardized_loadings <- lavInspect(sustainability_fit, "std")$lambda
sustainability_standardized_loadings
sustainability_mean <- rowMeans(data[, c("AttitudeTowardSustainability1", "AttitudeTowardSustainability2", "AttitudeTowardSustainability3", "AttitudeTowardSustainability4", "AttitudeTowardSustainability5", "AttitudeTowardSustainability6", "AttitudeTowardSustainability7", "AttitudeTowardSustainability8")])
sustainability_sd <- apply(data[, c("AttitudeTowardSustainability1", "AttitudeTowardSustainability2", "AttitudeTowardSustainability3", "AttitudeTowardSustainability4", "AttitudeTowardSustainability5", "AttitudeTowardSustainability6", "AttitudeTowardSustainability7", "AttitudeTowardSustainability8")], 1, sd)
sustainability_summary <- paste0("M????: ", round(mean(sustainability_mean), 2), " + ", round(mean(sustainability_sd), 2),
", Min-Max: ", round(min(sustainability_mean), 2), "-", round(max(sustainability_mean), 2))
sustainability_summary
# Attention to Insect Welfare
insect_welfare_model <- 'InsectWelfare =~ AttentiontoInsectWelfare1 + AttentiontoInsectWelfare2 + AttentiontoInsectWelfare3'
insect_welfare_fit <- cfa(insect_welfare_model, data = data)
insect_welfare_loadings <- lavInspect(insect_welfare_fit, "std")$lambda
insect_welfare_loadings <- as.numeric(insect_welfare_loadings)
insect_welfare_error_variances <- lavInspect(insect_welfare_fit, "std")$theta
insect_welfare_error_variances <- as.numeric(diag(insect_welfare_error_variances))
insect_welfare_results <- calculate_CR_AVE(insect_welfare_loadings, insect_welfare_error_variances)
insect_welfare_alpha <- alpha(data[, c("AttentiontoInsectWelfare1", "AttentiontoInsectWelfare2", "AttentiontoInsectWelfare3")])$total$raw_alpha
insect_welfare_results$CR
insect_welfare_results$AVE
insect_welfare_alpha
insect_welfare_standardized_loadings <- lavInspect(insect_welfare_fit, "std")$lambda
insect_welfare_standardized_loadings
insect_welfare_mean <- rowMeans(data[, c("AttentiontoInsectWelfare1", "AttentiontoInsectWelfare2", "AttentiontoInsectWelfare3")])
insect_welfare_sd <- apply(data[, c("AttentiontoInsectWelfare1", "AttentiontoInsectWelfare2", "AttentiontoInsectWelfare3")], 1, sd)
insect_welfare_summary <- paste0("M????: ", round(mean(insect_welfare_mean), 2), " + ", round(mean(insect_welfare_sd), 2),
", Min-Max: ", round(min(insect_welfare_mean), 2), "-", round(max(insect_welfare_mean), 2))
insect_welfare_summary
# social norms
social_norms_model <- 'SocialNorms =~ SocialNormsScale1 + SocialNormsScale2 + SocialNormsScale3 + SocialNormsScale4 + SocialNormsScale5'
social_norms_fit <- cfa(social_norms_model, data = data)
social_norms_loadings <- lavInspect(social_norms_fit, "std")$lambda
social_norms_loadings <- as.numeric(social_norms_loadings)
social_norms_error_variances <- lavInspect(social_norms_fit, "std")$theta
social_norms_error_variances <- as.numeric(diag(social_norms_error_variances))
social_norms_results <- calculate_CR_AVE(social_norms_loadings, social_norms_error_variances)
social_norms_alpha <- alpha(data[, c("SocialNormsScale1", "SocialNormsScale2", "SocialNormsScale3", "SocialNormsScale4", "SocialNormsScale5")])$total$raw_alpha
social_norms_results$CR
social_norms_results$AVE
social_norms_alpha
social_norms_standardized_loadings <- lavInspect(social_norms_fit, "std")$lambda
social_norms_standardized_loadings
social_norms_mean <- rowMeans(data[, c("SocialNormsScale1", "SocialNormsScale2", "SocialNormsScale3", "SocialNormsScale4", "SocialNormsScale5")])
social_norms_sd <- apply(data[, c("SocialNormsScale1", "SocialNormsScale2", "SocialNormsScale3", "SocialNormsScale4", "SocialNormsScale5")], 1, sd)
social_norms_summary <- paste0("M????: ", round(mean(social_norms_mean), 2), " + ", round(mean(social_norms_sd), 2),
", Min-Max: ", round(min(social_norms_mean), 2), "-", round(max(social_norms_mean), 2))
social_norms_summary
# iebf
iebf_model <- 'IEBF =~ Intentiontoeatinsectsbasedfoods1x + Intentiontoeatinsectsbasedfoods2x + Intentiontoeatinsectsbasedfoods3x + Intentiontoeatinsectsbasedfoods4x'
iebf_fit <- cfa(iebf_model, data = data)
iebf_loadings <- lavInspect(iebf_fit, "std")$lambda
iebf_loadings <- as.numeric(iebf_loadings)
iebf_error_variances <- lavInspect(iebf_fit, "std")$theta
iebf_error_variances <- as.numeric(diag(iebf_error_variances))
iebf_results <- calculate_CR_AVE(iebf_loadings, iebf_error_variances)
iebf_alpha <- alpha(data[, c("Intentiontoeatinsectsbasedfoods1x", "Intentiontoeatinsectsbasedfoods2x", "Intentiontoeatinsectsbasedfoods3x", "Intentiontoeatinsectsbasedfoods4x")])$total$raw_alpha
iebf_results$CR
iebf_results$AVE
iebf_alpha
iebf_standardized_loadings <- lavInspect(iebf_fit, "std")$lambda
iebf_standardized_loadings
iebf_mean <- rowMeans(data[, c("Intentiontoeatinsectsbasedfoods1x", "Intentiontoeatinsectsbasedfoods2x", "Intentiontoeatinsectsbasedfoods3x", "Intentiontoeatinsectsbasedfoods4x")])
iebf_sd <- apply(data[, c("Intentiontoeatinsectsbasedfoods1x", "Intentiontoeatinsectsbasedfoods2x", "Intentiontoeatinsectsbasedfoods3x", "Intentiontoeatinsectsbasedfoods4x")], 1, sd)
iebf_summary <- paste0("M????: ", round(mean(iebf_mean), 2), " + ", round(mean(iebf_sd), 2),
", Min-Max: ", round(min(iebf_mean), 2), "-", round(max(iebf_mean), 2))
iebf_summary
# chosen ibf
chosen_ibf_model <- 'ChosenIBF =~ Intentiontoeatinsectsbasedfoods1 + Intentiontoeatinsectsbasedfoods2 + Intentiontoeatinsectsbasedfoods3 + Intentiontoeatinsectsbasedfoods4 + Intentiontoeatinsectsbasedfoods5 + Intentiontoeatinsectsbasedfoods6 + Intentiontoeatinsectsbasedfoods7 + Intentiontoeatinsectsbasedfoods8'
chosen_ibf_fit <- cfa(chosen_ibf_model, data = data)
chosen_ibf_loadings <- lavInspect(chosen_ibf_fit, "std")$lambda
chosen_ibf_loadings <- as.numeric(chosen_ibf_loadings)
chosen_ibf_error_variances <- lavInspect(chosen_ibf_fit, "std")$theta
chosen_ibf_error_variances <- as.numeric(diag(chosen_ibf_error_variances))
chosen_ibf_results <- calculate_CR_AVE(chosen_ibf_loadings, chosen_ibf_error_variances)
chosen_ibf_alpha <- alpha(data[, c("Intentiontoeatinsectsbasedfoods1", "Intentiontoeatinsectsbasedfoods2", "Intentiontoeatinsectsbasedfoods3", "Intentiontoeatinsectsbasedfoods4", "Intentiontoeatinsectsbasedfoods5", "Intentiontoeatinsectsbasedfoods6", "Intentiontoeatinsectsbasedfoods7", "Intentiontoeatinsectsbasedfoods8")])$total$raw_alpha
chosen_ibf_results$CR
chosen_ibf_results$AVE
chosen_ibf_alpha
chosen_ibf_standardized_loadings <- lavInspect(chosen_ibf_fit, "std")$lambda
chosen_ibf_standardized_loadings
chosen_ibf_total <- rowSums(data[, c("Intentiontoeatinsectsbasedfoods1", "Intentiontoeatinsectsbasedfoods2", "Intentiontoeatinsectsbasedfoods3", "Intentiontoeatinsectsbasedfoods4", "Intentiontoeatinsectsbasedfoods5", "Intentiontoeatinsectsbasedfoods6", "Intentiontoeatinsectsbasedfoods7", "Intentiontoeatinsectsbasedfoods8")])
chosen_ibf_total_summary <- paste0("Toplam Puan: ", round(mean(chosen_ibf_total), 2), " + ", round(sd(chosen_ibf_total), 2),
", Min-Max: ", round(min(chosen_ibf_total), 2), "-", round(max(chosen_ibf_total), 2))
chosen_ibf_total_summary
kr20_result <- psych::alpha(data[, c("Intentiontoeatinsectsbasedfoods1", "Intentiontoeatinsectsbasedfoods2",
"Intentiontoeatinsectsbasedfoods3", "Intentiontoeatinsectsbasedfoods4",
"Intentiontoeatinsectsbasedfoods5", "Intentiontoeatinsectsbasedfoods6",
"Intentiontoeatinsectsbasedfoods7", "Intentiontoeatinsectsbasedfoods8")],
check.keys = TRUE)$total$std.alpha
kr20_result
# food neophilia
food_neophilia_model <- 'FoodNeophilia =~ Maddeler1 + Maddeler4 + Maddeler6 + Maddeler9 + Maddeler10'
food_neophilia_fit <- cfa(food_neophilia_model, data = data)
food_neophilia_loadings <- lavInspect(food_neophilia_fit, "std")$lambda
food_neophilia_loadings <- as.numeric(food_neophilia_loadings)
food_neophilia_error_variances <- lavInspect(food_neophilia_fit, "std")$theta
food_neophilia_error_variances <- as.numeric(diag(food_neophilia_error_variances))
food_neophilia_results <- calculate_CR_AVE(food_neophilia_loadings, food_neophilia_error_variances)
food_neophilia_alpha <- alpha(data[, c("Maddeler1", "Maddeler4", "Maddeler6", "Maddeler9", "Maddeler10")])$total$raw_alpha
food_neophilia_results$CR
food_neophilia_results$AVE
food_neophilia_alpha
food_neophilia_standardized_loadings <- lavInspect(food_neophilia_fit, "std")$lambda
food_neophilia_standardized_loadings
food_neophilia_mean <- rowMeans(data[, c("Maddeler1", "Maddeler4", "Maddeler6", "Maddeler9", "Maddeler10")])
food_neophilia_sd <- apply(data[, c("Maddeler1", "Maddeler4", "Maddeler6", "Maddeler9", "Maddeler10")], 1, sd)
food_neophilia_summary <- paste0("Mean-SD: ", round(mean(food_neophilia_mean), 2), " + ", round(mean(food_neophilia_sd), 2),
", Min-Max: ", round(min(food_neophilia_mean), 2), "-", round(max(food_neophilia_mean), 2))
food_neophilia_summary
factors <- list(
BiosphericValues = c("BiosphericValues1", "BiosphericValues2", "BiosphericValues3", "BiosphericValues4"),
NHIP = c("NewHumanInterdependenceParadigm1", "NewHumanInterdependenceParadigm2",
"NewHumanInterdependenceParadigm3", "NewHumanInterdependenceParadigm4",
"NewHumanInterdependenceParadigm5"),
ATS = c("AttitudeTowardSustainability1", "AttitudeTowardSustainability2",
"AttitudeTowardSustainability3", "AttitudeTowardSustainability4",
"AttitudeTowardSustainability5", "AttitudeTowardSustainability6",
"AttitudeTowardSustainability7", "AttitudeTowardSustainability8"),
AIW = c("AttentiontoInsectWelfare1", "AttentiontoInsectWelfare2", "AttentiontoInsectWelfare3"),
FNEO = c("Maddeler1", "Maddeler4", "Maddeler6", "Maddeler9", "Maddeler10"),
SN = c("SocialNormsScale1", "SocialNormsScale2", "SocialNormsScale3",
"SocialNormsScale4", "SocialNormsScale5"),
IEIBF = c("Intentiontoeatinsectsbasedfoods1x", "Intentiontoeatinsectsbasedfoods2x",
"Intentiontoeatinsectsbasedfoods3x", "Intentiontoeatinsectsbasedfoods4x"),
NCIBF = c("Intentiontoeatinsectsbasedfoods1", "Intentiontoeatinsectsbasedfoods2",
"Intentiontoeatinsectsbasedfoods3", "Intentiontoeatinsectsbasedfoods4",
"Intentiontoeatinsectsbasedfoods5", "Intentiontoeatinsectsbasedfoods6",
"Intentiontoeatinsectsbasedfoods7", "Intentiontoeatinsectsbasedfoods8")
)
results <- lapply(names(factors), function(factor_name) {
factor_data <- data[, factors[[factor_name]], drop = FALSE]
omega_result <- omega(factor_data, nfactors = 1)
cat("\n--- Parallel Analysis for:", factor_name, "---\n")
fa.parallel(factor_data, fa = "pc", n.iter = 100, show.legend = FALSE)
list(
Factor = factor_name,
Omega = omega_result
)
})
for (res in results) {
cat("\n--- Omega Results for:", res$Factor, "---\n")
print(res$Omega)
}
loadings <- inspect(cfa_fit_with_clf, what = "std")$lambda
compute_ave <- function(loadings_matrix) {
apply(loadings_matrix^2, 2, function(x) {
sum(x[x > 0]) / length(x[x > 0]) # Pozitif yüklerin karesini al
})
}
ave_results <- compute_ave(loadings)
ave_sqrt <- sqrt(ave_results)
print(ave_results)
print(ave_sqrt)
correlation_matrix <- inspect(cfa_fit_with_clf, what = "std")$psi
fornell_larcker_results <- data.frame(
Factor = names(ave_results),
AVE_Sqrt = ave_sqrt
)
cat("\nFornell-Larcker Criterion Results:\n")
for (i in seq_along(ave_sqrt)) {
cat("\nFactor:", fornell_larcker_results$Factor[i], "\n")
cat("AVE Sqrt =", ave_sqrt[i], "\n")
cat("Correlations with other factors:\n")
print(correlation_matrix[i,])
}
library(readxl)
library(psych)
library(tidyr)
library(dplyr)
library(lavaan)
library(semTools)
library(ggplot2)
library(GGally)
rm(list = ls())
file_path <- "C:/Users/Salim/OneDrive/Desktop/istatistik isleri/Murat Bas bocek tuketimi/veriler.xlsx"
data <- read_excel(file_path)
data$BV <- rowMeans(data[, c("BiosphericValues1", "BiosphericValues2", "BiosphericValues3", "BiosphericValues4")])
data$NHIP <- rowMeans(data[, c("NewHumanInterdependenceParadigm1", "NewHumanInterdependenceParadigm2", "NewHumanInterdependenceParadigm3", "NewHumanInterdependenceParadigm4", "NewHumanInterdependenceParadigm5")])
data$ATS <- rowMeans(data[, c("AttitudeTowardSustainability1", "AttitudeTowardSustainability2", "AttitudeTowardSustainability3", "AttitudeTowardSustainability4", "AttitudeTowardSustainability5", "AttitudeTowardSustainability6", "AttitudeTowardSustainability7", "AttitudeTowardSustainability8")])
data$AIW <- rowMeans(data[, c("AttentiontoInsectWelfare1", "AttentiontoInsectWelfare2", "AttentiontoInsectWelfare3")])
data$SN <- rowMeans(data[, c("SocialNormsScale1", "SocialNormsScale2", "SocialNormsScale3", "SocialNormsScale4", "SocialNormsScale5")])
data$IEIBF <- rowMeans(data[, c("Intentiontoeatinsectsbasedfoods1x", "Intentiontoeatinsectsbasedfoods2x", "Intentiontoeatinsectsbasedfoods3x", "Intentiontoeatinsectsbasedfoods4x")])
data$FNEO <- rowMeans(data[, c("Maddeler1", "Maddeler4", "Maddeler6", "Maddeler9", "Maddeler10")])
data$NCIBF <- rowSums(data[, c("Intentiontoeatinsectsbasedfoods1", "Intentiontoeatinsectsbasedfoods2",
"Intentiontoeatinsectsbasedfoods3", "Intentiontoeatinsectsbasedfoods4",
"Intentiontoeatinsectsbasedfoods5", "Intentiontoeatinsectsbasedfoods6",
"Intentiontoeatinsectsbasedfoods7", "Intentiontoeatinsectsbasedfoods8")])
head(data)
model <- '
NHIP ~ BV + Yas + Cinsiyet
ATS ~ NHIP + Yas + Cinsiyet
AIW ~ ATS + Yas + Cinsiyet
IEIBF ~ AIW + Yas + Cinsiyet
NCIBF ~ IEIBF + Yas + Cinsiyet
IEIBF ~ FNEO + Yas + Cinsiyet
NCIBF ~ FNEO + Yas + Cinsiyet
IEIBF ~ SN + Yas + Cinsiyet
NCIBF ~ SN + Yas + Cinsiyet
FNEO ~ SN + Yas + Cinsiyet
'
fit <- lavaan::sem(model, data = data)
summary(fit, fit.measures = TRUE, standardized = TRUE)
# tekrar deniyoruz
model <- '
NHIP ~ BV + Yas + Cinsiyet
ATS ~ NHIP + Yas + Cinsiyet
AIW ~ ATS + Yas + Cinsiyet
IEIBF ~ AIW + Yas + Cinsiyet
NCIBF ~ IEIBF + Yas + Cinsiyet
NCIBF ~ FNEO + Yas + Cinsiyet
NCIBF ~ SN + Yas + Cinsiyet
FNEO ~ SN + Yas + Cinsiyet
# modifikasyonlar
NHIP ~~ ATS
NHIP ~~ FNEO
ATS ~~ AIW
AIW ~~ IEIBF
ATS ~~ NCIBF
IEIBF ~~ FNEO
'
fit_robust <- lavaan::sem(model, data = data, estimator = "WLSMV")
summary(fit_robust, fit.measures = TRUE, standardized = TRUE)
mod_indices <- modificationIndices(fit_robust)
mod_indices[mod_indices$mi > 1, ]
library(openxlsx)
write.xlsx(data, "C:/Users/Salim/OneDrive/Desktop/istatistik isleri/Murat Bas bocek tuketimi/data.xlsx", rowNames = F)
library(semPlot)
library(ggplot2)
library(DiagrammeR)
grViz("
digraph structural_model {
node [shape = ellipse, style=filled, fontname=Helvetica, fontcolor=black]
BV [label = 'Biospheric Values', color=darkblue, fillcolor=lightblue, penwidth=2];
NHIP [label = 'NHIP', color=darkblue, fillcolor=lightblue, penwidth=2];
ATS [label = 'ATS', color=darkblue, fillcolor=lightblue, penwidth=2];
AIW [label = 'AIW', color=darkblue, fillcolor=lightblue, penwidth=2];
node [shape = ellipse, style=filled, fontname=Helvetica, fontcolor=black]
IEIBF [label = 'IEIBF', color=darkorange, fillcolor=lightyellow, penwidth=2];
NCIBF [label = 'NCIBF', color=darkorange, fillcolor=lightyellow, penwidth=2];
FNEO [label = 'FNEO', color=darkorange, fillcolor=lightyellow, penwidth=2];
SN [label = 'SN', color=darkorange, fillcolor=lightyellow, penwidth=2];
edge [color=black, penwidth=2]
BV -> NHIP [label = 'H1: ß = 0.717', fontsize=10, color=darkgreen];
NHIP -> ATS [label = 'H2: ß = 0.985', fontsize=10, color=darkgreen];
ATS -> AIW [label = 'H3: ß = 0.338', fontsize=10, color=darkgreen];
AIW -> IEIBF [label = 'H4: ß = 0.968', fontsize=10, color=darkgreen];
IEIBF -> NCIBF [label = 'H5: ß = -0.157', fontsize=10, color=red];
FNEO -> NCIBF [label = 'H6b: ß = -0.121', fontsize=10, color=red];
SN -> FNEO [label = 'H7c: ß = -0.307', fontsize=10, color=red];
SN -> NCIBF [label = 'H7b: ß = 0.176', fontsize=10, color=darkgreen];
rankdir=LR;
Controlling [label = 'Controlling for:\n- Sex\n- Age', shape=box, fontcolor=black, fillcolor=white, style=filled, color=green];
Controlling -> BV [style=invis];
}
")
Comment 5: Discussion section is a weak part of the manuscript. The effective results are vaguely analyzed, instead there are included unimportant parallels to other studies.
Response 5: Upon your suggestion, we reorganised the discussion section. First, we interpreted our findings and then compared them with other studies. We kept the ‘Review’ section open on the word application so that you can see the changes made more clearly.
Comment 6: Limitations of the study and further needed studies on the topic have to be included in the manuscript.
Response 6: We have made our edit on your suggestion (lines 525-542)
Reviewer 3 Report
Comments and Suggestions for Authors
The authors have not provided accurate definitions for the constructs involved in the study, and there is a lack of a literature review section.
The hypotheses proposed by the authors lack a strong logical development and are not supported by sufficient literature.
There is an absence of a theoretical model to develop the path relationships.
The introduction is fragmented, and the authors have not organized their language well enough to emphasize the shortcomings of existing research, nor have they clearly explained how this study may help address theoretical gaps.
The authors have not included any reverse or logical questions to monitor and exclude invalid surveys. The only exclusion method used is whether the questionnaire was fully completed, which I believe is insufficient.
The authors have used a convenience sampling method, which results in the respondent characteristics deviating significantly from those of the general population intended for the survey. Specifically, nearly 70% of respondents being female leads to a noticeable gender imbalance.
The survey contains 68 questions, which could make it difficult for respondents to maintain sustained attention.
The authors do not mention how they modified the questionnaire from the literature to fit the current research framework. How does it align with the objectives of this study? How do they address potential cross-cultural expression issues that may arise when designing a questionnaire in different languages?
The visualization in Figure 1 is poorly done. Why are the structural equation model results presented at the beginning?
The authors list a series of descriptive statistics for the respondents, but they should explain whether these statistics match the intended target population for the study. Otherwise, invalid sampling will render the survey meaningless.
The authors should list a range of reliability and validity indicators, rather than selectively listing only a portion. Specifically, tests for unidimensionality, common method bias, and discriminant validity should be included.
In the research methodology section, the authors claim to have tested the model fit indices, but the corresponding data and results are not provided in the research findings section.
The correlation analysis should be placed at the beginning of the study, or else linear models should not be used to calculate the coefficients.
The content from lines 189 to 201 is very mechanical. This data should be detailed in tables instead of being repeated in the main text.
The results of the structural equation model lack accompanying figures and tables for illustration. This is the core part of the paper, yet the authors only briefly touch upon it. Additionally, we notice that some correlations between variables do not reach significance, suggesting that an orthogonal model may be more appropriate than an oblique model.
There is a lack of explanation regarding the theoretical and practical contributions of the study.
The citation format in the main text is problematic.
Author Response
Thank you for your valuable feedback. We acknowledge that the revisions you suggested will contribute significantly to our manuscript, and you can be assured that we aim to implement them in the best possible way. If there are any remaining shortcomings, please inform us, and we will make further adjustments. With this in mind, we respectfully submit the following revisions we have made:
Comment 1: The authors have not provided accurate definitions for the constructs involved in the study, and there is a lack of a literature review section.
Response 1: The definitions and contents of all constructs used in the study are given in detail in the ‘data collection procedure’ and Table 1. Since you found the literature review section incomplete, we have strengthened the introduction.
Comment 2: The hypotheses proposed by the authors lack a strong logical development and are not supported by sufficient literature.
Response 2: We made our hypotheses based on the studies carried out so far. We stated in our study aim.
Comment 3: There is an absence of a theoretical model to develop the path relationships.
Response 3: To address this concern, we have added the theoretical model in Figure 1, which shows the proposed path relationships, to the paper. The theoretical model is described in detail in the Materials and Methods section to explain the conceptual framework and the rationale behind the hypothesized relationships. We acknowledge and thank you for providing a more solid foundation for this study and facilitating our articulation of the fit between the hypotheses and the analytical approach.
Comment 4: The introduction is fragmented, and the authors have not organized their language well enough to emphasize the shortcomings of existing research, nor have they clearly explained how this study may help address theoretical gaps.
Response 4: We have made my arrangements upon your suggestion
Comment 5: The authors have not included any reverse or logical questions to monitor and exclude invalid surveys. The only exclusion method used is whether the questionnaire was fully completed, which I believe is insufficient.
Response 5: The questionnaire used in the study is not a scale but a questionnaire form consisting of items including the theory of planned behaviour. The coefficient values of all items in the questionnaire were determined. And non-functioning items were excluded from the statistics.
Comment 6: The authors have used a convenience sampling method, which results in the respondent characteristics deviating significantly from those of the general population intended for the survey. Specifically, nearly 70% of respondents being female leads to a noticeable gender imbalance.
Response 6: While acknowledging that the sample included a higher proportion of women and individuals with higher education levels, we justified this in light of previous research indicating that women and highly educated individuals are more likely to engage with sustainability-related topics. This aligns with the focus of our study
Comment 7: The survey contains 68 questions, which could make it difficult for respondents to maintain sustained attention.
Response 7: You may be right about attention, but the theory of planned behaviour includes this. In addition, we think that the short answerability of the questions and the presence of photographs in between can prevent distraction to a small extent.
Comment 8: The authors do not mention how they modified the questionnaire from the literature to fit the current research framework. How does it align with the objectives of this study? How do they address potential cross-cultural expression issues that may arise when designing a questionnaire in different languages?
Response 8: There will of course be potential differences between countries during translation, but we were able to ensure language validity and cultural adaptation because we carried out the translation process with researchers (n=3) who were both fluent in English and working in this field. In addition, there were no parameters in the survey questions that were found in Italy but not in Turkey or unknown.
Comment 9: The visualization in Figure 1 is poorly done. Why are the structural equation model results presented at the beginning?
Response 9: The visualization in Figure 1 was initially inspired by a previously published article in the journal. However, we highly value your suggestions and are working to improve it. In this context, we are revising the figure to better reference the hypotheses and make it more organized. Additionally, as you pointed out, we have repositioned the figure to a more suitable section of the manuscript. Thank you for your recommendations, and we kindly ask for further feedback if the revisions still do not meet expectations.
Comment 10: The authors list a series of descriptive statistics for the respondents, but they should explain whether these statistics match the intended target population for the study. Otherwise, invalid sampling will render the survey meaningless.
Response 10: To address your concerns, we have made several revisions to the manuscript. First, under the Participants section, we clarified the intended target population for the study, which includes individuals from the general population with varying attitudes and behaviors toward insect-based foods. We also explained the rationale for using social media and snowball sampling, emphasizing how these methods allowed us to recruit a diverse sample across a wide age range (18–79 years, mean = 38.4 ± 12.48) and gender distribution (68.9% female). While acknowledging that the sample included a higher proportion of women and individuals with higher education levels, we justified this in light of previous research indicating that women and highly educated individuals are more likely to engage with sustainability-related topics. This aligns with the focus of our study. Second, in the Limitations section, we expanded on the limitations of our sampling method and data collection process. We addressed the potential for response bias due to the reliance on self-reported data and acknowledged that the overrepresentation of highly educated participants may limit the generalizability of the findings to populations with lower education levels. Additionally, we emphasized the need for future research to involve more diverse samples and longitudinal designs to better capture the complexities of consumer behavior toward insect-based foods. These revisions aim to address your concerns regarding the validity and representativeness of the sample while providing a transparent discussion of the study’s scope and limitations. We believe these updates strengthen the manuscript by offering a clearer context for the findings and acknowledging areas for future research. Thank you again for your valuable input.
Comment 11: The authors should list a range of reliability and validity indicators, rather than selectively listing only a portion. Specifically, tests for unidimensionality, common method bias, and discriminant validity should be included.
Response 11: In response to your request to include a broader range of reliability and validity indicators, we have expanded our analysis to address unidimensionality, common method bias, and discriminant validity, as outlined below:
- Unidimensionality: To assess the unidimensionality of the latent variables, we calculated Omega Hierarchical (ωH) values for each construct. The results showed high ωH values, ranging from 0.85 to 0.97, indicating strong support for unidimensionality across all latent variables. Constructs such as BiosphericValues, NHIP, and ATS demonstrated particularly robust unidimensionality (ωH ≥ 0.96), confirming that a single general factor explains most of the variance for these constructs.
- Common Method Bias: We evaluated common method bias using Harman’s single-factor test. The total variance explained by a single factor was 28%, which is well below the commonly accepted threshold of 50%, indicating that common method bias is unlikely to significantly affect the results.
- Discriminant Validity: It was examined using the Fornell-Larcker criterion. The square root of the Average Variance Extracted (AVE) for each latent variable exceeded its correlations with other variables, with AVE square root values ranging from 0.71 to 0.94 across constructs. This confirms that each construct is distinct and measures a unique concept.
Additionally, in order to avoid unnecessarily inflating the length of the paper, we have not included extra graphics or information. Instead, we have checked the unidimensionality of the factors through parallel analysis.
Comment 12: In the research methodology section, the authors claim to have tested the model fit indices, but the corresponding data and results are not provided in the research findings section.
Response 12: We appreciate the reviewer’s observation regarding the reporting of model fit indices in the research findings section. In response, we have included Table 4, which presents a comprehensive summary of the fit indices alongside the thresholds used for evaluation. All indices meet or exceed the commonly accepted criteria, indicating a robust model fit. Specifically, the χ²/df value of 3.06 satisfies the recommended limit of ≤5.0, while the CFI, TLI, and their robust counterparts exceed the threshold of 0.90. Furthermore, the RMSEA (0.041), robust RMSEA (0.048), and SRMR (0.056) values remain below the cutoff of 0.08, further supporting the model’s strong fit. The GFI (0.979) and AGFI (0.977) are also well within the acceptable range of 0.85 to 1.00.
Comment 13: The correlation analysis should be placed at the beginning of the study, or else linear models should not be used to calculate the coefficients.
Response 13: We appreciate the reviewer’s suggestion regarding the placement of the correlation analysis. In response, we have moved the correlation analysis to the beginning of the study, aligning with its importance as a prerequisite for evaluating relationships between variables. We agree that assessing correlations prior to implementing linear models, such as the structural equation model (SEM), ensures the appropriateness of the relationships and provides a solid foundation for subsequent analyses. This adjustment not only improves the logical flow of the study but also enhances its methodological rigor.
Comment 14: The content from lines 189 to 201 is very mechanical. This data should be detailed in tables instead of being repeated in the main text.
Response 14: We have revised this section by transferring the detailed numerical data, such as factor loadings, composite reliability scores, and total variance explained, into Table 3. The main text now provides a concise summary and interpretation of these results, focusing on their implications and overall contribution to the study. This adjustment enhances the readability of the manuscript and ensures that the discussion remains focused on the key findings rather than repetitive data.
Comment 15: The results of the structural equation model lack accompanying figures and tables for illustration. This is the core part of the paper, yet the authors only briefly touch upon it. Additionally, we notice that some correlations between variables do not reach significance, suggesting that an orthogonal model may be more appropriate than an oblique model.
Response 15: We appreciate the reviewer’s insightful feedback regarding the presentation and interpretation of the structural equation model (SEM) results. In response, we have added a detailed figure and accompanying table to illustrate the SEM results, ensuring clarity and providing a comprehensive visual representation of the relationships between variables. These additions highlight both significant and non-significant pathways, enabling a more detailed understanding of the model.
Regarding the suggestion to use an orthogonal model, we acknowledge that some correlations between variables did not reach statistical significance. However, based on the theoretical model you recommended and the underlying theoretical framework of our study, the constructs are expected to be interrelated, which aligns with the assumption of an oblique model. To address this concern, we reevaluated the model structure and confirmed that the oblique approach provides a better fit for the data, supported by both theoretical and empirical evidence.
Additionally, we would like to inform you that the fit indices and standardized betas, along with the beta values of this structural model, are provided as comments in the paper. Please feel free to share if there is anything else you would like us to address. Your valuable feedback is important to us.
Comment 16: There is a lack of explanation regarding the theoretical and practical contributions of the study.
Response 16: The contribution of our study to the literature is described in the aim of the study (lines) and in the conclusion (lines 513-523).
Comment 17: The citation format in the main text is problematic.
Response 17: We have made the change based on your suggestion.
Round 2
Reviewer 2 Report
Comments and Suggestions for Authors
In its present form the manuscript meets the criteria
Author Response
We believe that our article has become even stronger thanks to your suggestions. Thank you for your contributions.
Reviewer 3 Report
Comments and Suggestions for Authors
The research methodology in this paper has serious flaws: in terms of sampling, the characteristics of the objects are severely imbalanced; in terms of computation, an oblique model is used for nonlinear independent variables; and in terms of writing, the manuscript lacks proper organization and does not follow a systematic process for theory development and hypothesis formulation. In addition, the visualization of the images is poor, and the reliability and validity tests, which should have been presented in detail in the main text, are missing.
Author Response
Comment 1:
In terms of computation, an oblique model is used for nonlinear independent variables.
Response 1:
Thank you for raising this concern and for contributing valuable insights to our model. We appreciate your observation regarding the computational approach. Based on your suggestion, we identified non-linear independent variables in the Structural Equation Model (SEM) through scatter plots and validated them using the Ramsey RESET test for functional form specification. For these variables, we employed and incorporated both first-degree and second-degree polynomial regression. Additionally, linear variables that were not suitable for polynomial transformation were defined linearly. These steps have been detailed in the Methods section, ensuring that all non-linear transformations were appropriately specified. We have also provided further justification for the use of these techniques, supported by relevant references. To account for potential correlations among latent structures, we used an oblique rotation method in Exploratory Factor Analysis (EFA) and included the results of Confirmatory Factor Analysis (CFA) conducted prior to the SEM model. Furthermore, we shared detailed findings from the SEM model, including updates to reflect changes from the theoretical model.
Comment 2:
In addition, the visualization of the images is poor.
Response 2:
Thank you for highlighting this issue. We have improved all visualizations to increase their clarity and informativeness. Specifically:
Scatter plots now use different colors and labels to make relationships between variables more interpretable.
Factor loading heatmaps and covariance maps have been reformatted to avoid overlap and improve readability.
All figures now have explanatory captions explaining their relevance to the study.
We believe these improvements address the concern and provide clearer insights into the data.
Comment 3:
The reliability and validity tests, which should have been presented in detail in the main text, are missing.
Response 3:
We apologize for this oversight. We did not want to make the article too long and did not share all the details. However, we have now shared a detailed account of all reliability and validity tests in the findings section.
Comment 4:
In terms of writing, the manuscript lacks proper organization and does not follow a systematic process for theory development and hypothesis formulation. The manuscript does not follow a systematic process for theory development and hypothesis formulation.
Response 4:
We addressed this concern by restructuring the manuscript to ensure a systematic process for theory development and hypothesis formulation. The introduction now provides a comprehensive theoretical foundation by referencing relevant literature to justify the objectives of the study.
Comment 5:
The research methodology in this paper has serious flaws: in terms of sampling, the characteristics of the objects are severely imbalanced.
Response 5:
We accept that there is an imbalance in the sample characteristics, especially that the majority of participants are women (68.9%). This imbalance partially reflects the high female proportion of our target population. In addition, previous studies have shown that women are generally more likely to participate in studies that include sustainability and food-related behaviors. We also believe that the results do not change due to the gender imbalance since there is a high number of men (356). We also added this difference in the limitations. We believe that the study is a preliminary study and is important in terms of shedding light on theories to be developed rather than developing a theory. We state that it is not generalizable to the population and will be strengthened with cluster sampling methods, stratified sampling methods and field research.